# Emerging Trends in Nano-Driven Immunotherapy for Treatment of Cancer

**DOI:** 10.3390/vaccines11020458

**Published:** 2023-02-16

**Authors:** Gayathri Kandasamy, Yugeshwaran Karuppasamy, Uma Maheswari Krishnan

**Affiliations:** 1School of Chemical & Biotechnology, SASTRA Deemed University, Thanjavur 613401, India; 2Centre for Nanotechnology & Advanced Biomaterials (CeNTAB), SASTRA Deemed University, Thanjavur 613401, India; 3School of Arts, Sciences, Humanities & Education (SASHE), SASTRA Deemed University, Thanjavur 613401, India

**Keywords:** cancer therapy, nanoparticles, immunotherapy, tumour microenvironment, tumour recurrence

## Abstract

Despite advancements in the development of anticancer medications and therapies, cancer still has the greatest fatality rate due to a dismal prognosis. Traditional cancer therapies include chemotherapy, radiotherapy, and targeted therapy. The conventional treatments have a number of shortcomings, such as a lack of selectivity, non-specific cytotoxicity, suboptimal drug delivery to tumour locations, and multi-drug resistance, which results in a less potent/ineffective therapeutic outcome. Cancer immunotherapy is an emerging and promising strategy to elicit a pronounced immune response against cancer. Immunotherapy stimulates the immune system with cancer-specific antigens or immune checkpoint inhibitors to overcome the immune suppressive tumour microenvironment and kill the cancer cells. However, delivery of the antigen or immune checkpoint inhibitors and activation of the immune response need to circumvent the issues pertaining to short lifetimes and effect times, as well as adverse effects associated with off-targeting, suboptimal, or hyperactivation of the immune system. Additional challenges posed by the tumour suppressive microenvironment are less tumour immunogenicity and the inhibition of effector T cells. The evolution of nanotechnology in recent years has paved the way for improving treatment efficacy by facilitating site-specific and sustained delivery of the therapeutic moiety to elicit a robust immune response. The amenability of nanoparticles towards surface functionalization and tuneable physicochemical properties, size, shape, and surfaces charge have been successfully harnessed for immunotherapy, as well as combination therapy, against cancer. In this review, we have summarized the recent advancements made in choosing different nanomaterial combinations and their modifications made to enable their interaction with different molecular and cellular targets for efficient immunotherapy. This review also highlights recent trends in immunotherapy strategies to be used independently, as well as in combination, for the destruction of cancer cells, as well as prevent metastasis and recurrence.

## 1. Introduction

Cancer immunotherapy has gained popularity in recent decades as a viable therapeutic option for cancer patients, with impressive clinical outcomes. The landscape of cancer treatment has transformed due to the remarkable performance of monoclonal antibodies as immune checkpoint inhibitors in preclinical and clinical trials and the subsequent approval of these molecules for cancer therapy by the Food & Drug Administration (FDA). However, only a small percentage of patients and indications have benefited from most cancer immunotherapies [1]. This is because of the short circulation time of immune checkpoint inhibitors and shorter effect time. Additionally, off-targeting also results in adverse effects associated with the administration of immune checkpoint inhibitors [2]. Multiple attempts have been made to improve the effectiveness of cancer immunotherapy utilizing combination treatments that include numerous immune checkpoint inhibitors in clinical settings. However, the total patient response rate remains below 30%. Therefore, there is a need to improve conventional cancer immunotherapy by the adoption of newer and more effective strategies [3]. The advent of nanomedicine has offered an effective alternative to overcome the limitations of conventional immunotherapy. Over the past three decades, a wide range of nanoparticle-based drug delivery systems have been developed extensively as vehicles for the targeted administration of anticancer agents, small interfering RNAs (si-RNAs), oligonucleotides, plasmids, cytokines, and antibodies [1]. Due to their preferential accumulation in tumours due to the enhanced permeability and retention (EPR) effect, nanocarriers have been extensively investigated for tumour-specific delivery of anticancer medicines. In addition to the EPR effect, nanoparticles can be modified with ligands to specifically bind to receptors overexpressed on the cancer cell surface for active internalization and delivery of the therapeutic cargo, thereby minimizing off-target effects with concomitant increase in the treatment efficacy [4]. These benefits can be harnessed for immunotherapy by employing nanoparticle-engaging delivery systems for immunotherapeutic compounds that target the immune system and/or cancer cells [5] There has been several reports that have demonstrated that the distribution of immunotherapeutic drugs into tumour tissues or lymph nodes was successfully improved by fine-tuning the sizes, shapes, surface charges, and hydrophobicity of nanoparticles [4]. Nanoparticles also offer the advantage of co-encapsulating multiple immunotherapeutic drugs for simultaneous delivery to the targeted sites, thereby facilitating multi-modal and more potent therapeutic activity. Controlled and stimuli-responsive drug release in response to complex and immune-suppressive tumour microenvironments has also been affected using nanoparticles containing immunotherapeutic agents [3]. In this review, we discuss the recent advances towards nanoparticle-mediated immune engaging systems as immunotherapeutics for enhancing therapeutic efficacy in cancer immunotherapy and also the modulation of immunosuppressive tumour microenvironment employing adjuvants, cytokines and cytokine-like immune modulator nanoparticles, immunogenic cell death-inducing cytotoxic nanoparticles and engineered cells of mammalian and microbial origin. The potential of nanoparticle-based delivery methods to overcome constraints encountered in present cancer immunotherapy is also highlighted. The present review includes science citation indexed articles published from the year 2019 till date in Science Direct, PubMed and Google Scholar on immunotherapy for cancer. The key focus of these articles is on strategies for the effective transfer of cancer antigens or adjuvants to immune cells, particularly APCs (antigen-presenting cells) for induction of an anticancer immune response, as well as for modifying the immune-suppressive microenvironment to trigger an immune response [2]. Articles that have discussed use of nanoparticulate systems, for triggering an immune response against cancer and the emerging trends in cancer treatment using immune-modifying nanoparticles have also been elaborated in this review article.

## 2. The Immune Suppressive Tumour Microenvironment—A Formidable Fortress

Cancer immunotherapy works on the principle of stimulating antitumour immunity in tumours that results in destruction of cancer cells and slowing their progression [6]. The process of tumour recognition and killing of cancer cells by the immune cells is however challenging because tumours have been found to alter their immune microenvironment to an immune suppressive milieu for its progression, immune escape and survival [3]. The complex and heterogeneous tumour microenvironment (TME) consists of cancerous and non-cancerous cells that include immune cells, stromal cells, tumour-associated fibroblasts (TAFs), myeloid-derived suppressor cells (MDSCs), tumour-associated macrophages (TAMs), dendritic cells (DCs), and regulatory T cells (T_reg_) with immunomodulatory activities, which helps in promoting the tumour immune-suppressive microenvironment [6,7]. The metabolic dysregulation of cancer cells has been implicated in the development of the immune suppressive TME that inhibits activation of tumour infiltrating cytotoxic CD8^+^ T cells and natural killer (NK) cells [6]. A key aspect that has been demonstrated in many studies is the suppression of inflammatory cytokines and associated signalling pathways. Several strategies are now being designed to reverse this phenomenon. It is likely that further understanding of the immune suppressive microenvironment and its mechanism of promoting tumour survival can provide additional targets for complete eradication of cancer by immunotherapy [7]. The TME contains a wide variety of non-cancerous cells that aid in the survival of cancer cells, as well as their progression from the proliferative phase to an invasive phase and gaining resistance against various cytotoxic agents. Several studies have focused on investigating the role of these cells in the growth, proliferation, and survival of tumour cells, as well as creation and maintenance of an immunosuppressive environment. These have revealed some interesting facets that have served as leads for designing nanoparticle-mediated immunotherapy strategies. 

The ability of the tumour to escape from immune surveillance through creation of an immunosuppressive microenvironment is achieved by the secretion of cytokines that attract T_reg_ lymphocytes, myeloid derived suppressor cells, and macrophage polarisation [8]. Tumour-associated macrophages (TAMs) are key players in cancer progression by promoting angiogenesis, fibrous stroma deposition and metastasis. They also inhibit the T-cell antitumour response that is responsible for the immune suppressive milieu in the tumour by increasing the VEGF (vascular endothelial growth factor) and TGF-β (transforming growth factor—beta) production [7]. TAMs are believed to originate from peripheral blood monocytes migrating into the tumour from the blood vessels. A heterogenous population of TAMs exist in the TME that were classified as classic (proinflammatory) M1 and alternative M2 macrophages (anti-inflammatory). M1 macrophages are responsible for acute inflammatory response and antimicrobial defence and are characterized by the secretion of proinflammatory cytokines such as the interleukins IL18, IL12, IL6, and IL1β; chemokines such as CCL2 and CCL5; and TNFα (tumour necrosis factor-alpha). In contrast, M2 macrophages contribute towards wound healing and inflammation and secrete anti-inflammatory cytokines IL10 and IL4, and the chemokines CCL17 and CCL22. These sub-populations originate from the primary macrophages M0 and are reprogrammed as reversible M1/M2 due to external stimuli [8]. CD86 and TLR4 (toll-like receptor 4) are the markers of M1 macrophages, while M2 macrophages are characterized by CD163 and CD206. The enzymes iNOS and arginase1 (Arg1) are associated with M1 and M2 macrophages respectively [9]. M1 macrophages are abundant in richly vascularized regions and serve as antigen-presenting cells and generate reactive oxygen and nitrogen species (ROS and RNS) apart from proinflammatory cytokines, that trigger a robust T-cell response against the tumour [8]. M2 macrophages suppress antitumour immunity by suppression of CD8^+^ T lymphocytes and NK cells through expression of arginase1, indolamine-2,3-dioxygenase1, PD ligands (PDL1 and PDL2), ligand to the B7H4 receptor, and HLA ligands (HLAC, E/G). The transformation of M1 to M2 phenotype is enhanced by hypoxic condition and acidosis nature of tumour. Hypoxic conditions lead to higher levels of VEGF, CXCL8, ET-2 that mediate the transformation to the M2 phenotype that enhances secretion of proteolytic enzymes, growth factors (EGF, PDGF, HGF, and bFGF) and proangiogenic factors (VEGF, TNFα, IL8, bFGF, and CSF1/MCSF) for vascularization of the tumour tissue [8]. M2 phenotype of macrophages has also been implicated in tumour relapses after chemotherapy. Recent research has revealed that the tumour microenvironment consists of macrophages of both phenotypes—a fact evidenced through co-expression of iNOS and arginase1 [9]. Therefore, the sensitivity or resistance to cancer therapy is dependent on the dominant phenotype of the macrophages in the TME. Regulated transformation of the functional state of macrophages from immunosuppressive M2 to proinflammatory M1 phenotype is considered a major strategy in cancer immunotherapy.

An abnormality encountered in the TME involves presence of deregulated myeloid lineages. High levels of GM-CSF, IL-6 and oxidative stress in the TME affect the monocytic and granulocytic lineages, thereby altering bone marrow myelopoiesis [2]. This results in a heterogenous myeloid-derived suppressor cell (MDSC) population that promotes an immunosuppressive microenvironment. Further, the deregulated MDSCs do not differentiate into activated, immunogenic myeloid dendritic cells and inflammatory macrophages, thereby inhibiting tumour antigen presentation [7]. Therefore, identification of factors that inhibit the DC differentiation and function against tumours could help in designing immunotherapy strategies for superior cancer treatment.

It is now recognized that “tumour immunoediting”, a process where the immune system acts as a selection factor to alter the cell composition of the TME, can either totally eradicate a tumour or significantly reduce its growth or favour its survival [10]. If tumour cells acquire mutations that render their immune systems insensitive, the growth of these “immunologically silent” cells will always result in the greatest amount of tumour. Therefore, to effectively annihilate cancer, the immune system should be primed to distinguish cancerous and healthy cells. Cancer cells differ from normal cells in that they express a variety of tumour-associated antigens (TAAs). Specialised TAAs such as proteins with altered post-translational modifications or encoded by chromosomal abnormalities, and mutant gene products have the capability to form entirely in cancer cells and are referred to as “neoantigens” [11]. In other words, neoantigens can be referred to as non-self-proteins that are produced by tumour cells. Alternately, normal self-proteins that include embryonic or cell-differentiation antigens, which are absent in adult, differentiated cell types or are only negligibly expressed, might become immunogenic TAAs if they are produced or overexpressed by cancer cells [12]. The TAA-derived peptides are processed for presentation to tumour-specific lymphocytes by the membrane-bound major histocompatibility complex (MHC) class I or class II molecules found in antigen presenting cells (APC). Adenosine triphosphate, F-actin filaments, deoxyribonucleic acid (DNA), and TAAs have all been implicated in the activation of APCs in cancer. IFN-γ and TNF-α are cytokines produced by activated TAA-specific CD4^+^ T cells that can both inhibit tumour survival and increase the expression of MHC class I molecules by the tumour cells, thereby enabling recognition by TAA-specific CD8^+^ cytotoxic T lymphocytes (CTLs) [8]. Unfortunately, the TAAs-targeted strategies have not been very successful in clinical trials due to the low immunogenicity of TAAs and the potential for self-tolerance. Additional factors include insufficient thymic depletion, peripheral tolerance of TAA-reactive T cells, reduced peripheral TAA expression, TAA-reactive T cells with low TCR binding affinity, or limited TAA expression pattern during development. Therefore, immunotherapy strategies based on nanotechnology focus on introduction of TAA-expressing factors or design approaches to activate APCs for inhibiting tumour growth and progression. Neoantigen-based strategies may be more effective in circumventing the immune tolerance and activation of tumour-specific T cells, as they are expressed only by tumour cells [3]. Hence, nanocarriers have been employed for codelivery of neoantigens and adjuvants to harness the therapeutic potential of neoantigen-based customized immunotherapy.

## 3. Nanoparticle-Mediated Immunotherapy in the Treatment of Cancer

Nanoparticles having sizes below 300 nm have been extensively explored as nanocarriers for delivering therapeutic cargo to desired sites. Nanoparticles can be synthesized from different materials—metals, ceramics, polymers, and composites, in a wide range of size and shapes. Biodegradable polymeric nanoparticles have been extensively investigated for drug delivery applications as they will not cause long-term complications because they will not remain in circulation post release of the therapeutic cargo. Spherical nanoparticles do not possess any orientation effects that will affect their mobility and permeation into cells and hence have been widely employed for drug delivery. Further, surface modification of the nanoparticles could confer additional properties such as an increased circulation time, site-specific delivery, and triggered release of the cargo [1]. Encapsulation of the therapeutic molecules into nanoparticles can improve their accumulation in the target site, overcome issues pertaining to solubility and bioavailability of the drug, mask undesirable interactions of the drug with the biological system, as well as decrease the effective therapeutic dose required by minimizing loss due to distribution, metabolization or untimely elimination [13]. In the context of immunotherapy, the use of nanocarriers to deliver immunomodulators to cancer cells and lymphatic organs, stimulate immune cells, boost T-cell expansion, alter TME, overcome pathophysiological barriers, and accumulate in myeloid tissues and other vascular compartments has shown promising results that has made nano-based therapeutic approaches appropriate for immunotherapeutic strategies [14]. Nanocarriers have the potential to overcome limitations of conventional immunotherapy such as poor circulation time, off-targeting and short-live therapeutic action of immunomodulators and has been the topic of intense research in recent decades [1,13].

The nanoparticles employed for cancer immunotherapy include lipid-based nanoparticles (liposomes, phospholipid micelles, and solid–lipid nanoparticles), polymeric nanocarriers (synthetic and natural polymers fabricated as nanospheres, nanocapsules, micelles, nanogels, and dendrimers), inorganic nanocarriers (metal/metal oxide nanoparticles, carbon-based nanoparticles, mesoporous silica, calcium phosphate, etc.) (Figure 1) [1,15]. Among these, the most extensively studied nanoparticles for cancer immunotherapy are polymer-based nanosystems [2]. Due to their properties such as biodegradability, biocompatibility, and nontoxic nature, the FDA has approved a variety of polymers, including poly(ethylene glycol), poly(lactide-co-glycolic acid), and chitosan, for the preparation of nanoparticle systems for effective cancer immunotherapy. All of these nanoparticles have shown promise for the treatment of cancer by precise delivery of antigens and supplements to the disease site leading to activation of the immune system [2]. 

## 4. Modulation of Tumour Immunity Using Nanoparticles

Nanoparticle based delivery enables delivery of a single molecule or multiple immunostimulatory agents, with both spatial and temporal control. It is now well-established that APCs must present sufficient antigens to T cells over an adequate duration to effectively prime and activate them [16]. MHC class I molecules can display endogenous cytosolic antigens or cross-present exogenous antigens. The exogenous route is essential for producing CD8^+^ CTL responses, especially for tumour antigens and therefore improved cross-presentation of exogenous antigens is considered an invaluable tactic for effective cancer immunotherapy [17]. Several studies have demonstrated that when antigens were covalently attached to organic or inorganic nanobeads, they served as strong immunogens generating both cellular 6nd humoral responses [18]. A typical example employed calcinetin expressing cancer cell membrane antigen fragments that were conjugated to the surface of poly(lactide-co-glycolic acid) nanoparticles encapsulating the adjuvant R897 for immunotherapy against breast cancer [19]. The antigen coated nanoparticles were internalized by DCs resulting in their activation, while the sustained release of the adjuvant activated the toll-like receptor 7, thereby amplifying the immune response. The treatment reduced the tumour load and maintained a tumour-inhibitory environment owing to the memory effect, thereby preventing further episodes of tumour recurrence. In an interesting approach, exogenous lysis of antigen proteins into peptide fragments was reported using trypsin to mimic the intracellular antigen presenting process [20]. The antigen fragments were conjugated to calcium phosphate nanoparticles leading to better immune response evidenced by the upregulation of TNF-α and IFN-γ levels in the cancer environment. In vivo studies also revealed a marked suppression of the tumour progression in mice models of melanoma, colorectal cancer, and breast cancer. The results reveal that use of resorbable calcium phosphate nanoparticles may be a safe and nontoxic route for immunotherapy. The clinical delivery of various immunotherapy modalities, each with a different response rate, is also possible through nanocarriers (Figure 2). Immunotherapy strategies have been classified broadly as active and passive methods. The active strategy involves targeting TAAs uniquely expressed or overexpressed on tumour surfaces with the host immune system, while passive immunotherapy boosts the immune system’s normal anticancer response using monoclonal antibodies, lymphocytes, and cytokines. A combination of active and passive immunotherapy is now being explored as personalized medicine that could be realized using nanoparticle mediated co-delivery. 

Apart from using combinations of different types of immunotherapies or as a monotherapy, combinations of immunotherapy with other therapeutic strategies have also been attempted to improve destruction of cancer cells, while simultaneously preventing invasion and recurrence of cancer [21]. Immune checkpoint blockade (ICB) therapy has been frequently studied in combination with photodynamic therapy (PDT) for treatment of different types of cancers. The effectiveness of synergistic photodynamic-immunotherapy is constrained by adverse events caused by ICB antibodies and ineffective photosensitizer administration. To overcome these limitations, a nanocomplex comprising the photosensitizer Ce6 linked with acid responsive phenyl boronic acid along with poly(ethyleneimine) conjugated anti-programmed death ligand 1 (aPDL1) denoted as NC@Ce6. The acid-responsive nanocomplex can be converted into smaller nanoparticles (~28 nm) with a cationic charge in the weakly acidic TME that favours enhanced tumour penetration of aPDL1 and Ce6. This combination increased intratumoral infiltration of different immune cells, especially CD8^+^ T lymphocytes, when evaluated using melanoma and breast cancer induced mice models [22]. Along similar lines, modulation of the macrophage polarisation was attempted using a self-assembled nanoparticle (PyroR) formed by the combination of the photosensitizer pyropheophorbide-a (Pyro) for photostimuli-responsive cytotoxicity and the TLR agonist resiquimod (R848) for altering polarisation (Figure 3). Both co-delivered molecules enhance the immunogenic cell death by promoting the maturation of dendritic cells and activation cytotoxic T lymphocytes (CTLs) resulting in better regression in the tumour volume and prevented metastasis when tested in mouse bearing breast cancer [23].
Figure 2Currently available immunotherapy techniques. (1) Monoclonal antibodies, (2) Adoptive immunity, (3) Vaccines, and (4) Dysregulated immune system. These approaches are useful in evoking an immune response against cancers [22]. CC BY 4.0.
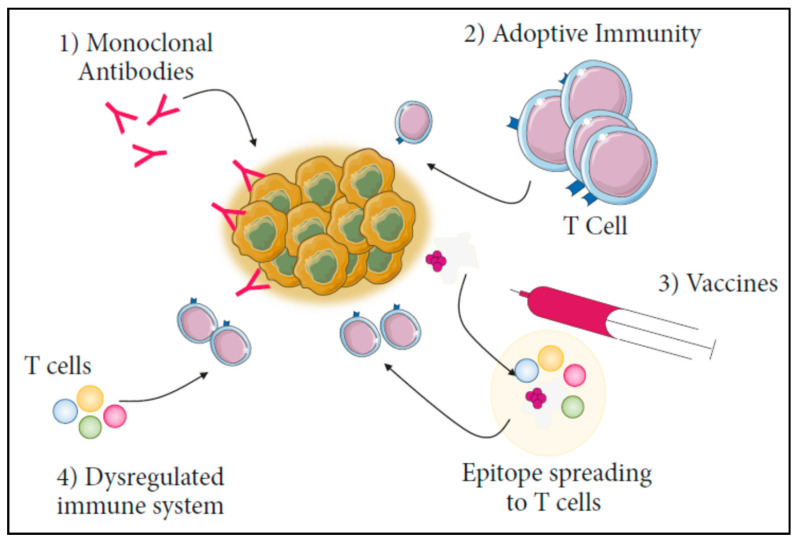

Figure 3Schematic depiction of the self-assembly of PyroR and its mechanism of photodynamic treatment (PDT) causing immunogenic cell death (ICD) and macrophage polarisation. Pyro and R848 could interact noncovalently between themselves to form PyroR. PyroR could inhibit initial tumour growth by PDT after passively accumulating at the tumour site and polarise macrophages from M2 to M1 phenotype to secrete cytokines. Further, the PDT aids immunotherapy by triggering immunological cascades, such as ICD activation, CRT and HMGB1 release, DC maturation, and lymph node migration. ICD cascades and M1 macrophage polarisation activate T cells to curb metastatic tumours [23]. Reproduced with permissions from Xiayun Chen et al., *Chemical Engineering Journal*, Elsevier, 2022.
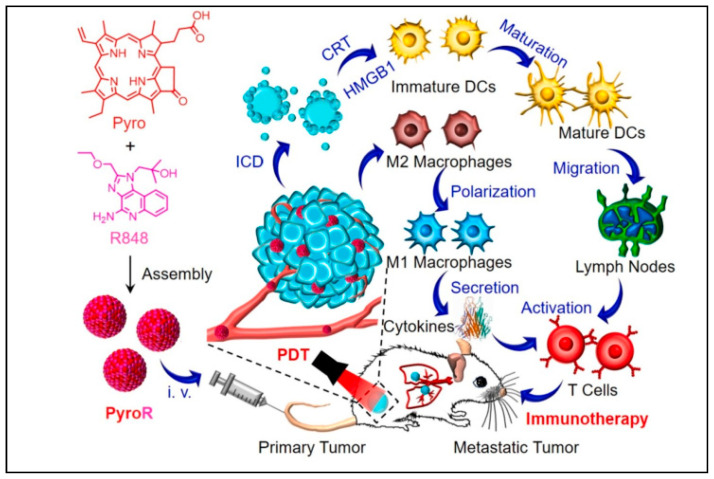


Multifunctional black phosphorus (BP) nanoparticles modified with PEGylated hyaluronic acid were employed for a combination of photothermal therapy (PTT), photodynamic therapy (PDT) and immunotherapy. The hyaluronic acid enabled internalization in CD44 expressing cancer cells, as well as transformed macrophages from M2 to M1 phenotype in the TME. The destruction of cancer cells due to PTT and PDT released TAAs that were captured and presented by the immune cells activated by hyaluronic acid (HA), thereby enabling a pronounced immune response and inhibition of recurrence due to memory effect. This nanoparticulate system showed promising effects both in vitro and in vivo models of breast cancer. This system caused immunogenic cell death (ICD) and released damage-associated molecular patterns (DAMPs) that accelerated DC maturation and activated effector cells for a robust antitumour immune response [24]. In another study Europium ions in combination with paclitaxel loaded in concave-shaped eudragit polymer nanoparticles were investigated successfully for chemoimmunotherapy by co-administering anti-PD1 antibodies using in vitro and in vivo models of colon carcinoma [25].

Hydrogel-based nanocarriers have also been successfully employed for combination therapy by serving as a matrix for sustained release of the immunomodulators. Mannose-OVA conjugates were prepared through chemical coupling with a 3:1 ratio of mannose to OVA. The conjugates were then loaded with rosuvastatin, an HMG CoA reductase inhibitor. This conjugate self-assembled to form Man-OVA-RSV nanoparticles that were loaded in gelatin hydrogel containing graphene oxide (GO) and metformin hydrochloride (MET) (Gel@NPs). The GO served as a photosensitizer that induced cell death through photothermal effect when illuminated with near infrared and also potentiated immunogenic cell death. The mannose served to specifically target the mannose receptors in the DCs, while the OVA activated DCs to serve as APCs to the antigens released by the cancer cells destroyed by photothermal effect. The rosuvastatin (RSV) enhanced the antigen presentation through inhibition of cholesterol biosynthesis and reduced the destruction of antigens, thereby prolonging the immune response. Thus, the multi-component Gel@NPs provided a combination of immunotherapy, photothermal effect and RSV-mediated metabolic reprogramming that effectively suppressed tumour growth by increasing proinflammatory cytokine levels due to activated immune cells. This strategy enhanced the efficacy of DC-mediated immunotherapy and the combination with checkpoint inhibition elicited strong antitumour immunity in melanoma-bearing mice model [26]. 

In another approach towards priming CD8^+^ T cells, poly (ethylene imine) (PEI)-based nanoparticle vaccine platform was developed to encapsulate CpG antigenic peptide and adjuvants. Co-administration of this system with STING (stimulator of interferon genes) agonist promoted better tumour infiltration and enhanced the antitumour efficacy through activation of immune response and concomitant infiltration of CTLs in MC-38 colon carcinoma and B16F10 melanoma murine models [27]. Dendritic cells and doxorubicin/CpG were co-loaded in a PEI-based hydrogel system that was fabricated for chemo-assisted immunotherapy against cancer. A significant benefit of the DC-based immunotherapy was that it overcame the therapeutic deficiency caused due to insufficient endogenous DC in vivo by introducing exogenous DCs. The CpG delivered by the hydrogel promoted the maturation and activation of DCs. Doxorubicin mediated cell death that further produced antigens that were presented by the DCs. Combining doxorubicin-stimulated tumour antigen presentation and delivery of DCs and immunomodulatory adjuvants, the hydrogel system provided a platform for chemo-assisted immunotherapy by inducing a potent CTL-mediated cytotoxicity towards cancer cells. Further, it significantly increased the infiltration of effector T cells, reduced the immunosuppressive microenvironment, and maximised the immune response when administered to melanoma induced mice [28]. 

The combination therapy involving FOLFOX (folinic acid (FnA), 5-fluorouracil (5-Fu), and oxaliplatin (OxP)) is limited by adverse effects and high drug resistance exhibited by the cancer cells. In an effort to overcome these limitations, the active cationic form of OxP ([Pt(DACH)(H_2_O)_2_]^2+^) and FnA were co-encapsulated in PEGylated lipid nanoparticles modified with aminoethyanisamide (nano-Folox), formed by nanoprecipitation. OxP was found to destroy cancer cells by recruiting immune components through mediating release of TAAs to activate the APCs. FnA was not cytotoxic but was found to sensitize the cancer cells to 5-Fu. The nanoparticles enhanced the internalization of FnA in the tumour cells therefore aiding the chemotherapy. The co-administration of 5-Fu with Nano-Folox resulted in superior tumour regression without any adverse effects in mice models of colorectal carcinoma (CRC) when compared to FOLFOX. The study also showed that Nano-Folox/5-Fu was improved by the anti-PD-L1 monoclonal antibody for a reduction in liver metastases in CRC-induced mice. These findings point to the potential of the combination of chemoimmunotherapy based on nano-Folox for the treatment of CRC [29]. It is also important to note that whether used as a monotherapy or in a combination, the effectiveness of immunotherapy depends significantly on the cancer type and stages, response rate, and expression of key biomarkers. Table 1 lists some of the recent studies employing nanoparticles for immunotherapy independently, as well as part of combination therapy with other cancer treatment strategies and their salient outcomes. 

## 5. Limitations in Using Nanoparticles for Conventional CANCER Immunotherapy

A wide range of nanoparticles have been explored for cancer immunotherapy. Despite their advantages, each type of nanoparticle possesses some limitations that hinder the achievement of complete cure from cancer. Micelles and liposomes formed from self-assembling amphipathic lipidic molecules are limited by their poor colloidal and structural stability, thereby increasing the risk of dose-dumping upon introduction into the biological system [75]. Hyper-branched dendrimeric systems though found extremely efficient in drug encapsulation, are limited by their cytotoxic nature owing to the high number of reactive functionalities at the terminus of the branches [76]. Polymersomes, which are self-assembled artificial vesicles formed using engineered amphiphilic block copolymers, are limited by their poor drug-loading efficiencies, high degree of serum instability, and inability to be synthesized below 100 nm sizes [15]. Biodegradable polymeric nanoparticles generally exhibit poor colloidal stability leading to agglomeration. It is also challenging to match the drug release profiles with their degradation profiles. Further, some of the by-products of their degradation could result in toxic effects [15] Non-degradable polymeric nanoparticles are not preferred for therapeutic applications owing to their poor ability to release the drug and possible adverse events arising due to their deposition in the biological tissues [76]. Inorganic nanoparticles, including metallic, semi-conductor, and ceramic nanostructures, are limited by their non-degradable nature, as well as possible adverse events associated with the products of their erosion or corrosion [77]. Additionally, the high surface area-to-volume ratios of the nanoparticles make them prone to opsonization and phagocytosis resulting in poor residence times within the biological system [78]. Another challenge in some of the nanocarriers, especially the reservoir type systems, is the burst release of the therapeutic cargo resulting in a sudden spike in their plasma concentrations, which may exceed the therapeutic limits [76]. 

Several strategies have been attempted to overcome these limitations of nanoparticles. Surface functionalization with poly(ethylene glycol) (PEG) chains denoted as PEGylation has been successfully employed to improve residence time of the nanoparticles and act as sites for conjugation of targeting ligands for site-specific delivery of the cargo or for linking antigen fragments to serve as artificial APCs [79]. PEGylated nanoparticles also exhibit superior colloidal stability, thereby overcoming the limitations presented by several nanoparticles [79]. PEGylation also retards burst release phenomenon owing to the fast dynamics of the PEG chains [80]. Dendrimeric systems have been modified with more benign and less reactive terminal groups to reduce their toxicity [15]. Most of the nanoparticulate systems currently being explored for immunotherapy are centered around biodegradable materials to avoid long-term challenges involved in their elimination. The nanoformulations intended for cancer immunotherapy must be recognized readily by the immune system components, unlike conventional drug delivery nanocarriers, which are designed to evade the immune system [81]. Therefore, a careful optimization of the degree of surface functionalities introduced is required. The search for newer materials and combinations for preparation of nanoparticles with superior characteristics remains an active area of research globally.

## 6. Recent Developments in Cancer Immunotherapy

Apart from delivery of antigens or immunomodulators, new directions are emerging for potentiating immunotherapy through nanoparticle mediated interventions. These include presentation of pathogen-derived biological components, use of engineered cells of mammalian or viral origin, exosome-based therapy, use of nucleic acids, employing gene editing or nano-optogenetic techniques, manipulating the mechanical properties of the TME and 3D printed scaffold-based strategies. These emerging strategies are discussed in the following sections. 

### 6.1. Microneedle-Based Immunotherapy

Microneedles are minimally invasive drug delivery systems that have micron scale diameter and lengths up to 1000 μm, which can penetrate into the epidermis and reach the dermis without affecting the blood capillaries or nerve endings, thereby facilitating painless transdermal delivery of the therapeutic cargo. Their ability to deliver cargo to the immune-rich environment of the dermis has triggered concerted efforts towards using this mode of delivery for immunotherapy applications [82]. Several types of microneedle types have been explored for transdermal delivery applications. These include removable solid microneedles that form micron-sized channels for permeation of the therapeutic agent through the stratum corneum (Figure 4). Hollow core solid microneedles remain on the skin and allow the diffusion of the therapeutic cargo into the epidermal and dermal layers. Coated microneedle arrays have a coating of the therapeutic agent that is introduced into the skin layers upon applying the patch. Dissolvable or degradable microneedle patches are not removable and release the drug through dissolution of the needles upon application on the skin surface. Hydrogel based microneedle patches enable controlled diffusion of the payload through volume changes associated with gelation and swelling in biological fluids. Non-degradable microneedles fabricated using silicon have not been preferred for immunotherapy owing to toxicity concerns, while stainless steel and other metal/alloy-based microneedle patches are not very compatible with delivery of hydrophilic moieties. Ceramic microneedles have been found to be unsuitable for loading thermosensitive molecules, while polymeric microneedles also pose challenges in maintaining their shape and mechanical integrity, as well as functional stability of the encapsulated molecules, especially those of biological origin [82]. Apart from the material chosen for fabricating microneedles, the length, tip diameter and array configuration are other parameters that will influence the therapeutic outcome. Dissolvable hydrogel microneedle array of 100 microneedles fabricated from poly(vinyl pyrrolidone) (PVP) with a tip-to-tip dimension of 1000 μm were loaded with nanoparticles prepared from the triblock copolymer F127 coated with cancer cell membrane components and loaded with the TLR9 agonist R837. This patch when administered to colon cancer induced mice, stimulated the skin residing antigen presenting cells that subsequently resulted in suppression of the tumour growth through elevated levels of TNF-α and IFN-γ [83]. Dissolvable hyaluronic acid microneedle patch was employed to deliver the octapeptide sequence SIINFEKL (OVA_257–264_), an epitope of the cytotoxic T-cells, that was conjugated to hyaluronic nanoparticles that effectively suppressed the growth and progression of melanoma in mice models by activating T-cell responses. The microneedle mediated delivery conferred longer residence time for the peptide when compared to the free peptide [84]. Synthetic nucleic acid adjuvants such as the TLR agonist polyriboinosinic:polyribocytidylic (poly(I:C)), a TLR3 agonist sequence have been successfully used to elicit a robust immune response upon loading in microneedle array patch fabricated using oligo sulfamethazine conjugated poly(β-aminoester urethane) (OSM-(PEG-PAEU)), which is a pH responsive polymer. The patch elicited excellent humoral and innate immune response when administered in mouse model of melanoma [85]. Chemo-immunotherapy was successfully implemented in melanoma bearing mice using a PVP microneedle patch loaded with lipid coated cisplatin, an antineoplastic agent and anti-PD1 antibodies for immunotherapy. The microneedle patch overcame the resistance exhibited by the tumour cells to free anti-PD1 due to the synergistic effect of the combination of cisplatin and anti-PD1 coated with lipids indicating a new direction in treatment of resistant forms of cancer [86].

Another interesting strategy employed pH responsive dextran nanoparticles containing anti-PD1 antibodies and glucose oxidase loaded in dissolvable hyaluronic acid microneedles. The glucose oxidase converted glucose in the biological fluids to gluconic acid that resulted in the reduction of pH, which, in turn, accelerated the swelling and degradation of the pH-responsive nanoparticles, leading to the release of anti-PD1. This strategy resulted in a significant increase in the magnitude of immune response when compared to conventional delivery strategies for anti-PD1. The same system was also used to elicit even better stimuli responsive immune response and tumour growth and recurrence control through co-delivery of the immunomodulator anti-CTLA4 along with anti-PD1 in mouse melanoma model [87]. This system could be explored further as a minimally invasive therapeutic option for treatment and management of cancer. Several other co-delivery strategies using microneedles have also been investigated with successful outcomes. Chitosan microneedle patch was co-loaded with mesoporous silica nanoparticles and similar nanoparticles formed from the core protein of Hepatitis B virus encapsulating the octapeptide SIINFEKL (OVA_257–264_). Another variant of the same system introduced CpG DNA adjuvant into mesoporous silica nanoparticles. The co-delivered adjuvants resulted in good control of tumour growth, progression, invasion, and recurrence in melanoma induced mice models due to T-cell stimulation and subsequent release of IFN-γ [88]. Another study had employed hyaluronic acid microneedles loaded with pH-responsive dextran nanoparticles co-encapsulating the hydrophobic photosensitizer zinc phthalocyanine and anti-CTLA4. The efficacy of this system was tested in mouse models of breast cancer where the combination of photodynamic therapy and immunotherapy mediated tumour cell kill, while the immune activation resulted in a memory effect preventing tumour recurrence [89]. Recently, micro-stereolithography technique was employed to fabricate a self-locking hyaluronic acid microneedle patch with precise geometry comprising a sharp tip for skin penetration, a wide body for skin locking and wings at the base for better skin insertion. The microneedles were loaded with anti-PD1 and SD-208, a potent inhibitor of TGF-β receptor type I (TβRI) kinase and the efficacy was demonstrated in a melanoma mice model [83]. In an interesting strategy, a novel rolling microneedle patch of stainless steel was fabricated for delivery of si-RNA against PD1 across a large cross-sectional area in the skin along with anti-PD1 antibodies (Figure 5). The microneedles also served as a microelectrode array that favoured transfection of the si-RNA upon application of voltage. The system was found to be safe when tested in normal and melanoma and colon cancer bearing mice [90]. The microneedles thus represent a minimally invasive strategy for delivery of adjuvants, as well as therapeutic molecules, for effective combinational immunotherapy. Some issues such as clogging and risk of infections owing to repeated applications still persist that may be overcome by intelligent design strategy and material choice for the fabrication of the microneedle array. 

### 6.2. Nucleic Acid-Mediated Immunotherapy

An important immunotherapeutic strategy involves activation of dendritic cells (DCs) loaded with tumour antigens and activated T lymphocytes from cancer patients. It has now been discovered that when compared to other types of antigens, especially proteins and peptides, use of mRNA encoding tumour antigens possess several advantages. These include their ability to be amplified from small amounts of tumour tissue, easily degradable nature without integrating into the host genome, thereby reducing risks, and the lack of a nuclear translocation requirement for its biological activity, thereby eliminating the challenge of nuclear delivery [91]. The recent global pandemic of SARS-nCoV-2 had renewed interest in mRNA-based vaccines and several pharmaceutical majors such as Pfizer, BioNTech, and Moderna have successfully launched mRNA-based vaccines to protect individuals from SARS-nCoV2 infection [91]. This has further spurred research towards development of mRNA-based immunotherapeutics for treatment of different malignancies. Numerous self-assembled nanostructures formed by different noncoding nucleic acids such as mi-RNAs, antisense aptamers, oligonucleotides, ribozymes, si-RNAs, CpG patterns, etc., with cationic matrices resulting in the formation of multifunctional nucleic acid nanoparticles (NANPs) have been successfully employed for regulation of diverse physiological processes. The innate nature of the immune system to recognize and eliminate foreign bodies, especially those that are charged similar to NANPs, can therefore be successfully exploited for cancer immunotherapy. Several nano-mediated nucleic acid delivery approaches have been reported for cancer therapy. A nanocomposite comprising hexapod-like structured DNA (hexapodna) along with CpG sequence and gold nanoparticles has been developed as an immunostimulatory DNA hydrogel. Hexapodna released after laser irradiation of the hydrogel effectively stimulated immune cells to generate proinflammatory cytokines. When injected intravenously into mice containing the EG7-OVA tumour followed by irradiation with a 780 nm laser, the gold nanoparticle-DNA hydrogel composite was found to elevate the tumour-associated antigen-specific IgG levels in the serum, as well as the local temperature in the tumour tissue. Consequently, there was an increased heat shock protein 70 (Hsp70) mRNA expression in the tumour tissue. The treatment further stimulated splenocytes to produce IFN-γ specific to the tumour-associated antigen. The study revealed that the therapy significantly slowed the growth of the tumour and increased the survival time in tumour-induced animal models clearly demonstrating the promise of photothermal cancer immunotherapy [92]. In a related study, Metal-X-Framework (MXF) of hafnium-CpG was used in colon cancer bearing mice for radioimmunotherapy. The Hf^4+^ responds to x-rays leading to cytotoxicity towards cancer cells. This releases TAAs, which are presented by APCs that infiltrate into the TME post-activation of DCs by the CpG sequence. The combinational radioimmunotherapy resulted in a pronounced inhibition of tumour growth, progression, invasiveness, and recurrence [93]. 

A DC-targeted nanovaccine platform was designed by incorporating functional DNA to cell membrane vesicles (CMVs) taken from tumour cells, CpG oligonucleotides, an agonist for toll-like receptor 9, and an aptamer that targets the DC SIGN receptor overexpressed on dendritic cells (DCs), thereby imparting target specificity. Experimental results revealed that these DNA-modified CMVs targeted DCs and accelerated their maturation. This nanovaccine platform displayed better therapeutic efficacy in the C57BL/6 and BalB/c that were subcutaneously injected B16-OVA cells and prevented recurrence of tumour due to the creation of memory effect. This study demonstrated that the combination of immune checkpoint inhibition with CMV-based nanovaccines may enhance treatment responses [94]. 

A pH-responsive interlocked DNA nanospring (iDNS) was employed to stimulate T cells in the low pH of the TME in vivo and reduce the adverse effects to the autoimmune system during immunotherapy. The interlocked structure of iDNS presented a stiffer substrate for ligand assembly when compared to double-stranded DNA, that aided superior spatial regulation of the ligand distribution. The study revealed that the acidic pH-driven assembly of iDNS aided in the control of CD3, the T-cell receptor distribution on the cell surface at the nanoscale. The weakly acidic tumour milieu (pH = 6.5) promoted conformation change of iDNS to a spring-like shrunken structure, which resulted in significant T-cell proliferation that curbed the growth, progression, and invasiveness of the cancer [95]. 

The self-assembling property of DNA sequences to form novel structures by DNA origami has been intelligently harnessed to serve as a nanocarrier co-encapsulating the immunomodulatory CpG and chemotherapeutic agent doxorubicin (Figure 6). The DNA nanocarrier was preferred due to its biocompatibility and low toxicity. The DNA nanocarrier was modified with AS1411 aptamer that targets nucleolin expressed on cancer cell surface for tumour-specific delivery. The encapsulated CpG was effectively taken up by macrophages and augments presentation of TAAs that elicited the release of cytokines and favoured infiltration of T lymphocytes. Interestingly, apart from enhancing the cytotoxicity of doxorubicin by promoting better internalization into tumour cells, the DNA tetrahedron carrier significantly reduced the adverse effects associated with doxorubicin administration due to non-specific accumulation in breast cancer bearing mice [96]. This seminal work paves way for further exploration of DNA nanocarriers for immunotherapy applications. Indeed, a recent study had employed DNA octahedron cages loaded with platinum nanoparticles and the immunomodulator R868 to reprogramme macrophages from M2 to M1 phenotype and reduced lung metastasis and inhibited recurrence in breast cancer induced mouse models [97].

In an innovative approach, artificial antigen-presenting cells (aAPCs) were prepared using lymphocytes by inserting cholesterol modified biotinylated DNA sequence in the cell membrane. Further introduction of avidin followed by introduction of OVA_257–264_(SIINFEKL) peptide-linked MHCs. This resulted in an artificial antigen-presenting cell (aAPC) with a controlled presentation of T-cell-activating ligands on the surface. When tested in melanoma-bearing mice, it was found that these lipid-coated lymphocytes not only displayed preferential migration to the tumour, but also augmented T-cell activation and proliferation of tumour-specific T cells. Further, when co-administered with aPD1, the surface-engineered aAPCs exhibited effective inhibition of tumour growth and reduced the mortality of the mice. Such surface engineered aAPCs could be explored further as an innovative platform for cancer immunotherapy applications in the near future [98]. 

### 6.3. Gene Editing Strategies in Immunotherapy

Gene editing based on Clustered regularly interspaced short palindromic repeat (CRISPR)-nuclease Cas9 ribonucleoprotein system has emerged as a front-runner in cancer therapeutic strategies, including immunotherapy. The system uses a single stranded RNA guide sequence to recognize and cleave the target DNA strands using Cas9 [99]. The cleaved strands when allowed to undergo repair may result in modifications such as insertion or deletion [100]. This importance of this versatile gene editing tool was highlighted by the conferment of the 2020 Chemistry Nobel Prize to its discoverers Emmanuelle Charpentier and Jennifer Doudna. To overcome the issues of delivering the CRISPR-Cas9 system to the cells of interest, most studies have employed polymeric nanocarriers. In a typical example, suppression of PD-L1 expression was attempted by knocking off of the cyclin-dependent kinase-5 (Cdk5) in melanoma cells through CRISPR-Cas9 delivered using the biodegradable cationic polymer poly(β-amino esters) (PBAE) [101]. The in vivo studies carried out in melanoma-induced mice revealed down-regulation of Cdk5 and PD-L1, thereby resulting in about 79% decrease in the tumour growth and inhibition of lung metastasis. The knockout also resulted in significant infiltration of CD4^+^ and CD8^+^ T lymphocytes in the tumour microenvironment indicating significant immune activation. In a similar strategy, PD-L1 was blocked by CRISPR-Cas9 derivatized with a low molecular weight branched poly(ethylene imine) [102]. The gene editing efficiency was confirmed by observing the down-regulation of PD-L1 and consequent suppression of tumour growth in melanoma bearing mice models. Virus-like nanoparticles have also been reported for delivery of CRISPR-Cas9 with singe stranded guide RNA specific for PD-L1 [103]. In an interesting variant, a nano-matryoshka type multi-stimuli responsive system was employed for delivering a CRISPR-Cas9 system for deleting genes encoding PD-L1 and protein tyrosine phosphatase N2 (PTPN2) [104]. Branched PEI derivative responsive to ROS was modified with PEG containing hyaluronic acid-RGD and MMP-recognition motif GPLGVRG. The first layer of the nanoparticle was cleaved in the tumour microenvironment rich in MMPs, while the second layer of the nanoparticle was lysed by hyaluronidase in the lysosome triggering its release in the cytosol. Finally, the ROS-rich intracellular milieu in the cancer released the CRISPR-Cas9 system for editing the genes encoding PD-L1 and PTPN2. The effectiveness of this strategy was proved by inhibition of the JAK/STAT pathway and stimulated a robust response from DCs and T cells, resulting in significant suppression of the tumour in tumour induced mice models. In another approach, CRISPR-Cas9 mediated down-regulation of PD-L1 was achieved in a ROS-dependent manner using a dendrimer system encapsulating the photosensitizer Chlorin-e6 [105]. Illumination of the photosensitizer triggered an increase an ROS that activated the editing action of CRISPR-Cas9 resulting in transformation of the tumour microenvironment to an immune supportive milieu. This caused maturation of DCs and enhanced infiltration of T cells, leading to immunogenic cancer cell death and inhibition of tumour metastasis. A similar strategy was employed using mitochondria targeting triphenylphosphonium (TPP) modified PEI containing cholrin-e6 and CRISPR-Cas9 for deleting PTPN2 and the tumour suppression efficacy through reversal of the immune suppressive environment was confirmed using melanoma bearing mice. The hyaluronic acid shell in the nanoparticle was hydrolysed by pre-administration of hyaluronidase for effective internalization and induction of oxidative stress through illumination [106]. Hyaluronic acid conjugated with tumour microenvironment sensitive peptides were incorporated in PEI and employed for simultaneously blocking CD47 and expressing IL-12 in tumour-associated macrophages through CRISPR-Cas9 editing, thereby reversing their tumour protective role and activating other immune components [107]. The efficacy of the editing approach was confirmed in vitro and in vivo models of melanoma. Recently, a cationic lipid based delivery of CRISPR-Cas9 approach was reported for deleting lactate dehydrogenase gene [108]. This resulted in an increased pH that served to activate T cells for producing proinflammatory cytokines. The efficiency of this strategy was evaluated in melanoma mice model.

Though gene therapy presents an exciting approach for immunotherapy, delivery of genes to specific sites remains a challenge despite the use of nanoparticles. Since oligonucleotides are negatively charged, cationic carriers have been chosen for forming an electrostatic complex with the oligonucleotide. However, careful control of the N/P ratios (positive to negative charges) is needed to ensure a subtle balance between complexation and release of the oligonucleotide at the desired site. Further, nucleus-specific delivery of genes overcoming the endo-lysosomal degradation presents another limiting factor in successful implementation of gene therapy. Recently, several stimuli responsive systems are being explored for effective site-specific delivery and release of the oligonucleotide from the nanocarriers and it may represent the next paradigm in genoimmunotherapy.

### 6.4. Exosome-Based Immunotherapy

Exosomes are cup-shaped vesicular structures about 30–150 nm in dimension secreted by cells into biological fluids for transport of miRNA, mRNA, peptides, and other cargo. This property has been harnessed for their applications in gene and drug delivery. Exosomes carry the signature of the cells from which they are secreted. Hence the exosomes from immune cells exhibit significantly high immunomodulatory properties [109]. Exosomes have been used to deliver mi-RNA such as miR155 that can stimulate and promote differentiation of macrophages to the proinflammatory M1 phenotype that can be used in cancer immunotherapy [109]. Alternately, DCs have been cultured in presence of cancer cells and the resultant exosomes were collected and used to stimulate the immune cells against the cancer cells in vivo for efficiently inhibiting the tumour progression [110]. Engineered exosomes can be designed to deliver drugs/cytokines/adjuvants to the tumour microenvironment while simultaneously activating the immune response against the cancer cells (Figure 7). 

The tumour derived exosomes (TEX) that naturally possess an immunosuppressive nature can be transformed into an immunostimulatory system through incorporation of activator ligands such as CpG, proinflammatory miRNA, or cytokines, or through the overexpression of heat-shock proteins HSP70 and HSP90 through thermal treatments. Alternately, genetic engineering of TEX for stimulating DCs or activating toll-like receptors could be carried out for effective cancer immunotherapy [111]. Exosomes harvested from blood leukocytes on a microfluidic platform were surface modified with melanoma derived peptides gp-100, MART-1 and MAGE-A3. These surface engineered exosomes were shown to enhance the proliferation of activated gp-100 targeted CD8^+^ cytotoxic T lymphocytes isolated from mice clearly demonstrating the benefits of exosome mediated immune activation for specific destruction of cancer cells [112]. This work for the first time employed a 3D printed microfluidic chip that could be a useful and convenient low-volume platform for harvesting and priming exosomes for cancer immunotherapy. Isolation and purification issues apart from challenges pertaining to effective loading of antigens and therapeutic agents still present a challenge in successful clinical translation of this promising approach. However, rapid advances in technology and improved understanding of the properties of exosomes may help eliminate the present limitations. 

### 6.5. Engineered Cells for Immunotherapy

Stem cell (SC) membranes have garnered attention in recent years as therapeutic carriers for targeting tissues or organs of interest because of their regenerative properties [113]. For instance, exosome membranes isolated from SCs have been employed for preparation of to increase the therapeutic potential of tailored drug delivery systems for immunotherapy. More recently, a novel coating method that employed a hybrid mixture of membranes from two different cells to form nanoparticles with better efficiency towards immunotherapy than conventional nanoparticles [113]. Cancer stem cells (CSCs) are a distinct subset of cell population found exclusively in tumours and possess properties resembling stem cells. They also exhibit distinct tumour characteristics such as resistance to drugs, invasiveness, and tendency to recur. These characteristics of CSCs are being harnessed for stimulating immune cells against cancer cells [114]. 

Stem cells can be engineered for serving as a reservoir for sustained generation of immune cells or tumour-specific effector cells for achieving remission. In addition, engineered stem cells are not patient specific and hence can be used as a therapeutic platform for different autologous cell therapies [115]. Poly(ethylene glycol)-dibenzocyclooctyne-pheophorbide (DPP) nanoconjugates have been successfully conjugated on the surface of human mesenchymal stem cells (hMSCs) (hMSC-DPP). The pheophorbide served as a photosensitizer that induced cell death, while the ability of hMSCs to migrate towards sites of inflammation was used for activating immune cells and concomitant secretion of proinflammatory cytokines such as hsp70, IL-2, IL-4, IL-6, IL-8, IL-12, granulocyte-macrophage colony-stimulating factor, and IFN-γ that direct the infiltration and accumulation of T cells, B cells, natural killer cells, and APCs. Treatment of breast cancer induced mice with hMSC-DPP reduced systemic immune-based responses and aided regression of the tumour. The PDT-mediated apoptosis further assisted in enhancing APCs, thereby serving as an effective platform for photoimmunotherapy [116].

Mesenchymal stem cells (MSC) from bone marrow localise in tumour tissues where they have been shown to support tumour growth and inhibit immune responses. In an effort to reverse this property and induce tumour regression, MSCs were modified to produce a TNF superfamily member homologous to lymphotoxin (LIGHT), which is also an immune stimulating factor (MSC-L). The results revealed a significant tumour-specific tropism of MSC-L both in vitro and in vivo. The system exhibited a robust immune response and promoted T-cell infiltration and caused tumour regression in mice models reversing the immunosuppressive environment [117]. 

### 6.6. CAR-T Therapy

Adoptive T-cell immunotherapy is rapidly gaining prominence in the present decade and involves isolation of patient T cells, activating them ex vivo and re-administration into the individual. Encapsulation of T-cell-stimulating agents or surface engineering of T cells with nanoparticles have also been attempted along with adoptive T-cell immunotherapy to overcome conventional limitations of T-cell therapy, such as intratumoral distribution, poor selectivity, and failure to circumvent the immunosuppressive TME [118]. Chimeric antigen receptor-T-cell (CAR-T) treatment is a cell engineering approach designed to enhance cytotoxic T-cell activation where a patient’s T cells are isolated, genetically altered with chimeric antigen receptors (CARs) employing a viral vector, and then reintroduced back into the same patient. CARs possess an intracellular signalling domain (e.g., CD3), and a linker to an extracellular antigen recognition domain, which is usually a fragment derived from the variable domain of an antibody. Upon recognition of a specific antigen by the extracellular domain, the signalling domain stimulates T-cell activation, proliferation, and cancer cell cytotoxicity. The US FDA had approved, in 2017, a CAR-T therapy for treatment of paediatric acute lymphoblastic leukaemia based on its impressive overall remission rate of 82.5% obtained from clinical trials. Currently, B-cell maturation antigen (BCMA) targeting Abecama^®^ for the treatment of refractory multiple myeloma and Carvykti™ for refractory multiple myeloma and CD19 targeting Breyanzi^®^ for large B-cell lymphoma, Kymriah™ for refractory diffuse large B-cell lymphoma, Tecartus™ for refractory mantle cell lymphoma, and Yescarta™ for high-grade, primary mediastinal and diffuse large B-cell lymphomas have received approval from FDA for cancer immunotherapy. However, the effectiveness of CAR-T therapy in treatment of haematological cancers and solid tumours is not very satisfactory. Additionally, there are manufacturing safety concerns indicating there is further scope for improvement of this platform for cancer therapy [119]. Engineering CAR-T cells using viral vectors have posed several concerns due to permanent CAR expression leading to undesirable consequences. Recently, ionizable lipid nanoparticles (LNPs) were developed to deliver mRNA to human T cells ex vivo. Seven formulations transfected using lipofectamine for enhanced mRNA distribution were shortlisted from a library containing 24 ionizable lipids. Among these candidates, the best-performing LNP formulation formed using C14—4, was used for delivery of CAR mRNA to primary human T cells. The results revealed negligible cytotoxicity and a comparable expression of CAR to electroporation. When co-cultured with Nalm-6 acute lymphoblastic leukaemia cells, CAR-T cells created using C14—4 LNP exposure exhibited significant cytotoxicity that were comparable to CAR-T cells generated by electroporation. These results suggest that LNPs could be a better option for mRNA-based CAR-T-cell engineering [120]. Stem cells have also been engineered to express different CARs or T- cell receptors (TCRs) against TAAs that can be invaluable in the treatment of solid tumours and blood cancers [121]. Recently, studies to non-invasively visualize the in vivo fate of CAR-T cells upon introduction into the circulation have been carried out using iron oxide nanoparticles that has also now obtained FDA approval [122]. 

### 6.7. Nano-Optogenetics for Immunotherapy

Optogenetics was first introduced in the mid-2000s, and it employs light to control functions of cells that are genetically engineered to be photoresponsive [123]. Though initially conceptualized to modulate neuronal functions, the field has expanded to encompass therapeutic approaches, including immunotherapy. Engineered T cells with a blue light driven melanopsin-inducible Ca^2+^ switch to trigger proinflammatory cytokines were successfully demonstrated to destroy HepG2 liver cancer cells upon illumination with blue light. Lentiviral vectors were employed in this study to introduce the gene of interest [124]. In a similar strategy, peri-operative immunotherapy was demonstrated at the site of melanoma resection in a mouse model using mesenchymal stem cells engineered to produce the cytokines IFN-β, TNF-α, and IL-12 upon illumination to far red light. The engineered cells were dispersed in a polysaccharide hydrogel matrix (Vitrogel^®^) for implantation at the resection site (Figure 8). The photoactivation of the engineered cells resulted in a memory effect that prevented recurrence of cancer at the site [125]. 

Introduction of plasmids in the target cancer cell with the gene of interest that encode for an antigen or cytokine whose expression is triggered by a photo-responsive promoter could be invaluable in the context of immunotherapy. In this context, nanoparticles for gene delivery have been employed. Since electromagnetic radiation with lower wavelength have poor penetration depth in biological tissues, upconversion nanoparticles have been employed to realize the therapeutic potential of optogenetics. These nanoparticles are excited in the longer wavelength and emit in the shorter wavelength. Lanthanides are generally explored for upconversion strategies as they get excited in the NIR region and can be tuned to emit in the blue region [126]. Light switchable CAR (LiCAR) containing T cells have been engineered to become photoresponsive by introducing a pair of photodimerisable regions whose dimerization will be necessary for activation of the T cell. Core-shell upconversion nanoparticles of β-NaYbF_4_:0.5% Tm@NaYF_4_ coated with silica were employed for activation of the photodimerization of the LiCAR containing engineered CD8^+^ T-cells. The system exhibited superior control of tumour progression without any off-targeting effects conventionally encountered with conventional CAR-T based immunotherapy [127]. Though the combination of nanoparticles and optogenetics offers an exciting approach for cancer treatment with precise spatial and temporal control, there are concerns on the possible toxicity of the lanthanide nanoparticles employed in the technique that need to be addressed before clinical translation [126]. 

### 6.8. Virus and Viral Components for Immunotherapy

The ability of viruses to activate immune cells has elicited interest in the research community for use in immunotherapy. Viruses have been effectively employed to stimulate cytokine production from dendritic cells through activation of the TLR-PAMP signalling cascades. Since viruses pose a risk of causing infections in humans, non-pathogenic viruses and plant viruses have been employed for this strategy [128]. For instance, the non-pathogenic Sendai virus has been used to activate DCs to produce IFN-β and IL-6, which, in turn, can activate the NK cells and macrophages [129]. In order to target the activated immune cells towards breast cancer cells, the study had employed anti-CD47 antibodies that will effectively inhibit the activation of the CD47- signal regulatory protein-α (SIRPα) pathway, which confers protection to the cancer cells against detection and elimination by immune cells. Since platelets and erythrocytes also exhibit CD47, the Sendai virus and anti-CD47 were packaged in PLGA nanoparticles for superior accumulation in the tumour microenvironment and to reduce non-specific disruption of the normal blood cells and platelets. The nanoparticles were also incorporated with a NIR fluorescent probe for visualizing their in vivo fate. The combined effect of DC stimulation by the virus and blockage of CD47-SIRPα signalling resulted in increased survival time in breast cancer induced mice from 16 days to 39 days and a significant reduction in the tumour volume. Interestingly, the treatment also conferred humoral immunity to the animals and prevented recurrence of the tumour. 

Many plant viruses have also been explored for cancer immunotherapy. Papaya mosaic virus (PapMV) treatment was found to activate CD8 positive cytotoxic T cells with a concomitant decrease in the myeloid-derived suppressor cells through inducing the generation of proinflammatory cytokines [130]. When combined with administration of anti-PD1 antibodies, the PapMV treatment resulted in better control of melanoma progression in syngenic mice models, clearly demonstrating the potential of the virus to act as an adjuvant for immunotherapy. Cowpea mosaic virus (CPMV) and its nucleic acid free counterpart eCPMV containing only the capsid were both demonstrated to stimulate immune cells in syngenic mouse models of melanoma [131]. The study also showed that the modification of the capsid with poly(ethylene glycol) (PEG) chains did not interfere with the magnitude of immune activation. A unique aspect of CPMV was that it targeted immune cells rather than the cancer cells making it more effective for immunotherapy. Along similar lines, CPMV conjugated with the cost-effective anti-PD1 peptide sequence SNTSESF was found to elicit robust immune response and improved antitumour efficacy in syngenic mice with ovarian cancer [132]. The peptide conjugation overcame the immunosuppressive environment in the cancer and fine-tuning the conjugation density could lead to further improvement in the therapeutic outcome. Recently, a dissolvable poly(vinyl pyrrolidone) microneedle array containing 225 microneedles with magnesium nanoparticles loaded with CPMV were successfully employed for reducing tumour growth and recurrence in melanoma bearing mice models through stimulating the innate immune response [133]. This strategy represents the beginning of minimally invasive and self-administered therapeutic options for cancer treatment and management. 

Tobacco mosaic virus (TMV) with its characteristic nanotube structure was effectively employed for conjugation of the TLR agonist (2-methoxyethoxy-8-oxo9-(4-carboxy benzyl)adenine (IV209) [134]. The surface of TMV-IV209 nanoconjugate was coated with the photoresponsive polydopamine for NIR-mediated photothermal annihilation of cancer cells in mice models of melanoma. The survival, as well as tumour regression, was superior in the TMV-IV209 treated groups subjected to NIR irradiation. Interestingly, similar outcome was also observed in the only laser-treated animals. However, the rate of tumour recurrence was much lower in the TMV-IV209-laser groups, which demonstrates that the viral-mediated immunotherapy also imparts a “memory effect” that protects the animals from future episodes of tumour recurrence (Figure 9). Other plant-derived viruses, such as Sesbania mosaic virus, Tomato bushy stunt virus, Red clover necrotic mosaic virus, Potato virus X, etc., have been explored for anticancer drug delivery and may find use as adjuvants in immunotherapy regimens in the near future [128]. 

Bacteriophages that belong to the class of prokaryotic viruses have also been used to trigger immune responses against cancer as they are generally regarded safe for eukaryotic systems. A filamentous bacteriophage M13 was successfully coated with the cationic polyethylene imine (PEI), which was further adsorbed with cancer antigens (peptides, cancer-derived membrane components, proteins, etc.) and administered in melanoma and breast cancer-induced orthotopic mice models [135] (Figure 10). The results revealed a strong activation of antigen presenting cells and when co-administered with αPD-1-mediated immune checkpoint inhibition therapy, the cancer regression efficiency was superior. Additionally, the recurrence of cancer was reduced, indicating the promise of this strategy for cancer therapy. 

### 6.9. Oncolytic Virotherapy

Oncolytic viruses are genetically engineered viruses that can selectively infect and proliferate only in cancer cells. Since the introduction of the concept in 1991, oncolytic virotherapy has picked pace with initiation of Phase I clinical trials in 1998. In 2015, the first oncolytic virotherapy using Talimogene laherparepvec (T-Vec), a double mutated herpes simplex virus, was approved by FDA for treatment of melanoma [136]. Apart from their ability to induce apoptosis of the host cancer cells, oncolytic viruses also activate the immune cells and also lead to neutrophil repolarization and clumping in the blood vessels leading to destruction of the tumour vasculature [137]. A naturally occurring reovirus has been found to replicate only in cells with activated Ras signalling and one of its variants reolysin has been explored for cancer therapy. Several clinical trials using engineered and natural oncolytic viruses are underway for immunotherapy of cancers. These include coksackie virus for bladder cancer and acute myeloid leukaemia, an attenuated measles virus for leukaemia and a gamma herpes virus for HIV-induced lymphoma. In order to prevent early recognition and elimination of the oncolytic viruses, as well as to ensure better tropism towards cancer cells, encapsulation of the viruses within membrane vesicles, albumin, complexation with magnetic nanoparticles, manganese/calcium carbonates, or mesenchymal stem cells, have been attempted. Enhanced therapeutic outcomes have been demonstrated in cancer models treated with a combination of oncolytic viruses and immune checkpoint inhibitors such as anti-PD1 or cytotoxic CAR-T cells [137]. For instance, an oncolytic adenovirus carrying CpG islands was coated by extrusion with cancer cell membrane from lung cancer or melanoma (ExtraCRAd), depending upon the animal model chosen [138]. In all cases, the membrane wrapped oncolytic virus exhibited superior retardation of the tumour growth and improved survival rates (Figure 11). The membrane-wrapped virus outperformed the naked virus clearly demonstrating the merits of superior internalization of the membrane wrapped system through pathways other than receptor mediated endocytosis. Further, the coating minimized the levels of neutralizing antibodies, thereby extending the lifetime of the virus in the system. Despite concerns of possible risk-causing mutations and scale-up issues involving generation of engineered viruses, it is evident that such engineered viruses in combination with other forms of cancer therapy could be explored for treatment of aggressive and drug resistant cancers using appropriate models to usher in a new generation of cancer therapeutics. 

### 6.10. Bacterial Immunotherapeutics

Ever since the seminal effort of Willian Coley in administering heat inactivated Streptococcus and Serratia marcescens for cancer therapy, the field of bacterial immunotherapeutics has experienced a significant growth in the recent decades. The bacterial membrane contains abundant polysaccharides and glycolipids that could serve as recognition motifs for the lectin receptors present in immune cells. They have been found to bind to toll-like receptors triggering proinflammatory cytokines. Further, anaerobic bacteria such as *Escherichia coli* and *Salmonella typhimurium* exhibit selective accumulation and proliferation in the hypoxic environment of solid tumours, thereby making them effective agents for immunotherapy, as well as for the delivery of immune checkpoint inhibitors or chemotherapeutic agents. They could also be engineered to deliver plasmids encoding suicide genes for cancer therapy. Bacterial membranes have also been known to secrete vesicles that have been explored for delivery of anticancer therapeutics and also elicit a robust immune response for highly efficient cancer annihilation [139]. Engineered Salmonella has been used for delivering plasmid containing a gene encoding for an antigen to cancer through the oral route. Though cost-effective and simple, the therapeutic efficiency of this system is limited due to inactivation of the bacteria in the highly acidic gastric fluid. Additionally, the rapid phagocytosis of the Salmonella reduces the efficiency of immune stimulation. To overcome these drawbacks, a blend of beta cyclodextrin and poly(ethylene imine) were complexed with DNA encoding for GFP through electrostatic interactions [140]. This polyplex was assembled over the Salmonella membrane and the system demonstrated good cell internalization and immune activation. This trend was also observed when orally administered in melanoma-bearing mice. However, there still exist safety concerns over the use of engineered bacteria and their cost-effectiveness for large-scale clinical implementation. Nevertheless, this stratagem could be further explored for other types of cancer using robust animal models. Additionally, development of methodologies for large-scale engineering of the bacteria for immunotherapy could bring down the cost in the near future before being tested for possible clinical translation.

In an interesting approach involving a combination of engineered bacteria and nanoparticles, *E. coli* was engineered to express a L-arabinose dependent suicide gene upon stimulation with blue light [141]. Upconverting nanoparticles NaYF_4_:Yb/Tm@NaGdF_4_:Yb coated with polyethylene imine and conjugated with folic acid for tumour specific internalization were prepared. When co-administered in breast cancer induced mice, selective accumulation of the *E. coli* and folic acid tagged nanoparticles were observed. The upconverting nanoparticles when irradiated with NIR emitted blue light activated the promoter in the engineered bacteria. This in turn triggered the expression of the suicide gene in the animals that were fed with arabinose (Figure 12). The results revealed good tumour inhibition by the combination of engineered bacteria and upconverting nanoparticles. Similarly, engineered *E. coli* triggered by blue light to release TRAIL was successfully demonstrated when used in combination with folic acid tagged upconversion nanoparticles of NaYF_4_:Er,Yb@NaYF_4_ or C@CaF in colon cancer induced mice showed significant abrogation of tumour [142]. This optogenetic strategy could serve as a platform technology for other forms of cancer and delivery of cancer-specific gene constructs.

Both Gram positive and Gram-negative bacteria produce outer membrane vesicles (OMVs) that comprise the signature biomolecules of the bacteria and contain nucleic acids, toxins, or peptides/proteins as the cargo. These OMVs range between 20 and 400 nm in diameter and have been found to trigger the immune response by the pattern recognition receptors present in the immune cells [143]. Since OMVs are non-replicating, they present a safer choice over live bacteria for immunotherapy. The vesicles could be used to deliver other therapeutic agents, thereby making them useful agents for combination therapy. OMVs derived from Salmonella were coated with distearoyl phosphatidylethanolamine linked with poly(ethylene glycol) and the integrin recognition tripeptide motif RGD. The coated OMVs were then coated over pluronic F127 micelles containing the anticancer prodrug Tegafur [144]. When administered in melanoma-bearing mice, the OMV-coated nanoparticles showed improved survival, reduced proliferation and inhibition of metastasis. Additionally, the therapeutic utility of tegafur was enhanced by delivery through OMV-coated micelles through better internalization, as well as sensitization of the cancer cells by the immune activation. Similarly, gold nanoparticles coated with OMVs derived from *E. coli* were successfully employed to destroy glioblastoma cells by a combination of radiotherapy and immunotherapy [145]. Along similar lines, OMVs from *E. coli* were coated over copper sulphide nanoparticles for successful photothermal and immunotherapy against breast cancer induced mice [146]. The CuS-OMV combination also repolarized tumour-associated macrophages and induced maturation of DCs resulting in better inhibition of tumour proliferation and metastasis (Figure 13).

### 6.11. Fungal Derivatives as Immunotherapeutics

Pathogen-associated molecular patterns (PAMPs) in microbial membrane surface can be used effectively to activate toll-like receptors. This is the underlying concept in employing fungal membrane polysaccharides for immunotherapy. Fungal beta glucans are abundantly present in the fungal cell wall and are primarily composed of glucose units linked via α- or β-glycosidic linkages. Both 1,3 and 1,6 linkages are found in these molecules that can be linear or most often branched structures. The glucans activate toll-like receptors by binding with lectin receptors such as dectin-1 and CR3 receptors expressed in macrophages and DCs. This is evidenced through increase in proinflammatory cytokine levels mediated through NF-κB signalling. Several beta glucan extracts from mushrooms, including *Ganoderma lucidum*, *Polyporus rhinoceros*, *Agaricus blazei*, *Sparassis crispa*, *Grifola frondose*, *Coriolus versicolor*, *Inonotus obliquus*, *Pleurotus ostreatus*, *Phellinus linteus*, and *Pleurotus pulmonarius*, have been shown to strongly stimulate macrophages and DCs. This property has been employed to retard colon cancer proliferation both in vitro, as well as in vivo [147]. Fungal beta glucans have also been employed as carriers of anticancer drugs, thereby facilitating superior cancer destruction through a combination of chemo and immunotherapy. However, most of the studies have employed beta-glucan extracts that also contain impurities such as proteins and other polysaccharides. Attempts to purify beta-glucans conserved their dectin-1 binding ability but destroyed their TLR activating property, thereby rendering them ineffective for immunotherapy [148]. To overcome this limitation, six purified water-soluble acidic polysaccharides from the fungus *Inonotus obliquus* were evaluated for their immunotherapy potencies. It was found that they elicited strong responses from macrophages in wild-type, as well as TLR-4 knockout mouse macrophages indicating their TLR4 independent mechanism. Molecular studies revealed that these polysaccharides had the ability to interact with different pattern recognition receptors, thereby making them highly effective immune activators [148]. Another attempt employed a synthetic beta glucan that activated macrophages through TLR4 and dectin-1 resulting in an increase in the proinflammatory cytokines with a simultaneous reduction in the anti-inflammatory cytokines [147]. Fungal mannans have also been explored for activating T_H_17 cells that triggered natural killer and cytotoxic T cells. The immunotherapy was further potentiated by employing mannan conjugated nanoparticles modified with anti-OX40 antibodies that serve as an agonist of the co-stimulatory receptor OX40 [149].

### 6.12. Herbal Interventions for Immunotherapy

Recently, there has been a renewed interest in exploring phytochemicals and plant-derived formulations for therapeutic applications. In the context of immunotherapy, there has been several attempts to harness the potential of phytochemicals and their derivatives to elicit immune response from the immune cells. A polysaccharide derived from a popular Chinese medicinal plant *Angelica sinensis* were chemically conjugated with MMP recognition peptide motifs and the chemotherapeutic drug doxorubicin [150]. The conjugate self-assembled into a nanoparticle that accumulated in the tumour through enhanced permeation and retention. The lysis of the peptide bonds by the MMPs in the tumour microenvironment results in release of the drug and the polysaccharide. The polysaccharide activates the immune cells, thereby augmenting the cytotoxic effects of doxorubicin that was demonstrated by elevated cytokine levels in mice administered with the nanoparticles. The phytomolecule served as both a carrier, as well as an immune stimulator. Further assessment of this system in tumour models may help in establishing the therapeutic potential of this phytomolecule. Several studies have shown that compounds such as emodin, anemoside A3, dihydroartemisinin, dihydroisotanshinone I, gastrodin, puerarin, ginkgolide B, salidroside, etc. that are present in different herbs used in traditional Chinese medicine have been shown to modulate immune cell responses in different cancer models both in vitro and in vivo [151].

Similarly, herbal formulations such as Biejiajian pill, Yupingfeng, Baiying extract, Shuangshen granules, Xiaoyaosan, Yi-yi-fu-zi-bai-jiang-san, etc. have also shown promising immunomodulatory effects that could be harnessed for cancer immunotherapy [151]. To overcome bioavailability and targeting issues, nanoparticle mediated delivery of these phytocompounds have also been attempted. Liposomal formulations of tanshinone, an active ingredient from the Chinese herb Danshen, along with doxorubicin were prepared and surface modified with prostate specific membrane antigen for site-specific delivery in cancer tissue. However, evaluation of its potential for immunotherapy is yet to be reported. In another study, a lecithin containing nanoemulsion of puerarin, a polyphenol present in kudzu root used in Chinese traditional medicine was found to reverse the immunosuppressive tumour microenvironment and promote infiltration of T cells [62]. The nanoemulsion also deactivated tumour associate fibroblasts, reduced the ROS and facilitated the cytotoxic action of paclitaxel in triple negative breast cancer mouse models. Another study had employed mesoporous silica nanoparticles for co-encapsulating Astragaloside III isolated from an herb *Astragalus membranaceus* commonly used in Chinese medicine along with the photosensitizer chlorin-e6 [152]. The system activated natural killer cells and increased levels of proinflammatory cytokines. This system shows promise for cancer treatment with a combination of photodynamic therapy and immunotherapy and significantly reduced tumour volume when evaluated in colorectal cancer mice model. *Lycium barbarum* polysaccharides extracted from the medicinal herb *Lycium barbarum* were demonstrated to enhance DC activation, infiltration and antigen presentation resulting in a pronounced immune response [153]. Several herbal polysaccharides have been investigated for their ability to activate immune response both in the free state, as well as in the encapsulated form [154]. Though the preliminary studies on these molecules and formulations show promise for cancer immunotherapy, there are concerns about their toxicity that need to be addressed through systematic studies [151].

### 6.13. Regulating 3D Matrix Architecture for Immunotherapy

Three-dimensional scaffolds have garnered interest in the context of cancer immunotherapy because they offer additional advantages of retention of immune cells at the implanted site, facilitate the controlled release of cytokines, adjuvants, immune checkpoint inhibitors, serve as antigen presenting matrices, aid T-cell expansion, infiltration, trafficking, priming and modulation [155]. However, selection of an appropriate material chemistry and topography is influenced by many factors. Experiments have revealed that the activation, polarization, cytotoxicity, migration, deformation and recognition of immune cells and cancer tissue comprising fibroblasts, tumour associated macrophages and the cancer cells, are all strongly influenced by the mechanical, topographical and chemical properties of the scaffold [156]. Matrix stiffness remodelling through the disruption of matrix protein cross-links, destruction of metastatic niche, and inactivated tumour-associated fibroblasts has been proposed as a potential strategy to enhance the effectiveness of chemotherapy and immunotherapy (Figure 14).

Polymeric micelles formed by self-assembly of the co-polymer PEG-b-poly(benzyl-L-glutamate) were employed to deliver tranilast, an antifibrotic drug to breast cancer bearing mice [157]. The micellar delivery enhanced better internalization of the drug in the tumour microenvironment that targeted the tumour associated fibroblasts significantly reducing its numbers. Consequently, increased T-cell infiltration along with a significant reduction in the mechanical stiffness of the tumour extracellular matrix was observed. This strategy resulted in a marked reduction in the tumour progression and lung metastases, as well as inhibition of tumour recurrence. In a more elaborate study, the combination of liposomal doxorubicin and tranilast were found to reduce tumour growth and invasiveness in breast cancer models [158]. The combination reduced mechanical stiffness, thereby relieving the stress on the tumour vasculature and normalizing the blood flow to the tumour cells. This abrogated the hypoxic conditions that improved the cytotoxic efficiency of doxorubicin. The ability to control tumour progression by targeting tumour associated fibroblasts and reducing matrix stiffness is gaining momentum and more classes of mechanotherapeutics are being explored for cancer immunotherapy. Matrix degrading enzymes such as hyaluronidase when delivered in combination with anticancer drugs such as doxorubicin and PD-L1 silencing RNA sequence were found to effective in chemoimmunotherapy of cancer [159]. This system was found to improve T-cell infiltration through matrix degradation, activate T cells through PD-L1 regulation and destruct cancer cells by the co-delivered cytotoxic drug. The reduction in matrix stiffness also contributed to better retention of the drugs in the tumour apart from overcoming hypoxia and immunosuppressive microenvironment. Research in this field has ushered in a new era of “mechanogenetics”, where the modulation of mechanosensitive molecule Piezo signalling in immune cells is being attempted to reduce the expansion of myeloid-derived suppressor cells through combination of nanotechnology and genetic reprogramming [156]. Better understanding of this facet can bring in an additional factor to improve the immunotherapy efficiency against cancer. Recently, use of single cell-RNA sequencing has gained pace for understanding gene-level modulations brought about by immunotherapeutics [160]. This can aid in tailoring therapeutic strategies to optimally regulate the tumour microenvironment and activation of the immune components. Along similar lines, proteomics has been employed to understand the molecular mechanism of tumour suppression brought about by the 2D carbon nanomaterial graphdiyne [161]. Apart from repolarizing the macrophages from the M2 to M1 phenotype, graphdiyne was found to modulate the MAPK and TLR signalling pathways along with inlterleukin-1 processing, thereby activating T cells in a partially STAT3 independent route. A recent study had focussed on understanding the cancer genes that determine the responsiveness or resistance of the cancer cells towards PD-L1 inhibitors [162]. A panel of 10 genes were identified across different types of cancers that were then used to screen various therapeutic agents and nanoparticles. This led to the identification of the best nano combinations that could be employed for eliciting optimal immune response to abrogate cancer growth and progression paving way for personalized immunotherapy.

Several 3D scaffold geometries have been explored for cancer immunotherapy. These include implantable, injectable hydrogel scaffolds, and microneedle patches. Each category has its own merits and limitations. Implantable scaffolds enable the controlled release of adjuvants and facilitates activation of immune cells but is an invasive strategy with risk of poor biocompatibility and/or mechanical failure. Injectable scaffolds are flexible in shape and minimally invasive while favouring the controlled release of adjuvants. However, gelation properties, mechanical failure, and poor biocompatibility issues limit their use. Transdermal microneedles are minimally invasive and favour sustained release but are highly localized and restricted to superficial layers [155]. Alginate, fibrin, methacrylated hyaluronic acid, poly(lactic acid), poly(ethylene glycol)-modified poly(caprolactone), pluronics, and self-assembling RADA16 peptide gels have been explored for fabrication of 3D scaffolds for immunotherapy applications [155]. Recently, a 3D-printed hydrogel scaffold of poly(ethylene glycol) containing heparin for retaining CCL21 cytokine was employed for rapid expansion of T cells isolated from a patient as part of adoptive T-cell therapy. This 3D printing enabled personalized scaffold development, while the use of the scaffold itself overcame current limitations of slow expansion of the primed T cells [163]. This strategy shows promise and may herald in an era of personalized immunotherapy in the near future. A 3D artificial athymic organoid model has been developed for differentiation and expansion of pluripotent stem cells to different T cell lineages that are capable of secreting a wide range of cytokines for immunotherapy [164]. Similar attempts to generate patient-derived tumour infiltrating lymphocytes are also being made using tumour organoids. A recent study has postulated that 3D oriented porous scaffolds could aid in expansion of a large number of activated immune cells that could auger well for cancer immunotherapy [165]. However, it is evident that discovery of newer material combinations with tailored properties coupled with systematic studies with larger sample sets are needed to establish the therapeutic utility of such systems.

Table 2 summarizes the salient aspects of some the emerging strategies discussed in the above sections for the immunotherapy of cancer.

## 7. Nanoparticles with Immunotherapy in Clinical Trials

CAR-T, T cell receptors, tumour infiltrating lymphocytes, and natural killer (NK) cells are among the adoptive cell therapies being developed for a variety of reasons. New marketing techniques are also being used. Antagonizing antibodies such as PD-1 and CTLA-4 are currently in the lead in the immunotherapy market. It is evident that antagonists by themselves, however, do not always result in the majority of patients responding well. The major issue with CAR-T technology involves relapse reported in individuals with CD19-negative paediatric B-cell acute lymphoblastic leukaemia (B-ALL). Similarly, CD22-directed CAR-T fails in individuals with CD19 naive or resistant B-ALL and low CD22 expression. Additionally, there have been some concerns about possible resistance being developed against CAR-T cell immunotherapy for leukaemia and lymphoma. Active research into checkpoint inhibitors, adoptive T-cell treatment, oncolytic viruses or strategies involving modification of the tumour microenvironment with different combinations of nanoparticles and/or adjuvants can lead to an improved therapeutic outcome using immunotherapy [168]. Technology for innovative gene editing combined with knowledge of cancer biology could maximise the effectiveness of chimeric antigen receptors T-cell (CAR-T) technology for different types of cancers.

Nanoplatforms have demonstrated exceptional qualities, including high loading capacity, variable porosity, and ability to be targeted to the desired site, thereby greatly increasing the efficiency of immunotherapy while minimising its harmful and adverse effects. Development of nanoparticles for immunotherapy has undergone significant advances, but their use in clinical studies for cancer immunotherapy is still in its early stages. Several clinical trials are now underway since the past decade to ascertain the usefulness of nanoparticle-based immunotherapy independently or in combination with conventional immunotherapy to understand their efficacy in the treatment of cancer. A Phase I clinical trial is currently underway using crystalline Hafnium oxide nanoparticles that exhibit immune stimulating effects in addition to its radio-enhancing properties along with anti-PD1 for treatment of individuals suffering from any form of primary cancer and with lung or liver metastasis [169]. Another Phase I trial used nanoliposomes to deliver the microRNA-34a (miR-34a) mimic to suppress immunosuppressive tumour genes in individuals with solid tumours. The initial results in a cohort of 47 individuals with advanced solid tumours pre-treated with dexamethasone were encouraging though additional studies are required to ascertain the tolerability of the formulation [170]. Another Phase I trial involved administration of a lipid-coated mRNA-4157 that encodes for a wide range of tumour antigens to elicit tumour responses independently, as well as in combination with the anti-PD1 humanized antibody pembrolizumab. The initial study showed good dose tolerance paving way for Phase II trials for this combination [171]. Another Phase I trial had investigated the anticancer potential of a RNA-lipoplex for triggering dendritic cell maturation and T-cell response in a small cohort of patients with advanced melanoma. The results indicated that all treated individuals exhibited robust T cell responses that need to be further confirmed with additional stringent clinical trials [172]. Table 3 lists some of the trials involving nanoparticles and immunotherapy for treatment of different types of cancer.

## 8. Concluding Remarks

In this article, the effects of changing the tumour microenvironment, immunotherapeutic drugs, and immunomodulatory substances, as well as cancer immunotherapy, were discussed. To safely and effectively control the immune system in cancer patients, nanoparticulate immunotherapies can be highly challenging to translate into the clinic. The first issue is that in vitro experiments do not accurately represent in vivo circumstances since they lack crucial elements such as host and tumour-derived microenvironmental variables, although maybe being effective for intracellular evaluation. Animal models can also be a valuable source of in vivo data, but their capacity to simulate the extraordinarily complicated processes of human carcinogenesis, physiology, and progression is constrained. High specificity, long-term efficacy, and bioavailability of payloads might be clinically challenging for delivery system design and optimization in order to assure low systemic cytotoxicity.

Immunotherapy has extensively employed nanoparticles to deliver immunomodulatory agents or serve as artificial antigen presenting cells or stimulate the maturation and secretion of cytokines by immune cells or to reprogramme the tumour associated macrophages from the immunosuppressive M2 phenotype to proinflammatory M1 phenotype. A wide range of nanomaterials from synthetic to natural biomaterial-based have been explored for improving anticancer immunity. Engineering cells and employment of gene editing tools have also gained traction as cancer immunotherapeutics. Most studies have clearly demonstrated that use of nanoparticles enhance the magnitude and duration of immune response when compared to free immunotherapeutic agents. Nanoparticles also have been surface modified to enhance localization in the tumour microenvironment, thereby preventing adverse events associated with off-targeting and improving the therapeutic outcome. The attractiveness of nanoparticles is in their adaptability and functionalization, which enables their design in a range of forms, sizes, and functions to satisfy varied requirements. The nanoparticles are delivered to tumour tissues with precision, and they exploit the hyperpermeability of the tumour vasculature, either directly or indirectly, enhancing cancer immunotherapy or reducing side effects of chemotherapeutic drugs. A number of in vitro and in vivo studies have demonstrated positive outcomes using nanoparticles in cancer immunotherapy. These include significant drug protection against degradation, intracellular delivery, regulated and sustained release, and the prevention of multi-drug resistance in different types of cancers. Although still in its infancy, the manipulation of the immune system by nanoparticle-based immune-modulating medicines offers intriguing cancer treatment options. Numerous studies have used nanoparticles as delivery systems for antigens and costimulatory molecules to be co-ordinately delivered to APCs, resulting in improved CD^4+^ and CD^8+^ T responses against tumour. The effectiveness of nanoparticulate APCs designed to directly activate T cells has also been established. By using small-molecule inhibitors and gene-loaded nanocarriers to deplete/repolarize TAMs, MDSCs, and Tregs, as well as by blocking immunosuppressive factors in tumours, armies of immune cells against tumours can be strengthened. Several Phase I clinical trials involving nanoparticles have been initiated in the past decade indicating the growing importance of this strategy in cancer immunotherapy. Emerging paradigms such as optogenetics, stimuli responsive systems, 3D printed scaffolds and modulation of the tumour microenvironment promise to transform the landscape of immunotherapy. One of the unique advantages of immunotherapy is its ability to prevent tumour recurrence and metastasis that can enhance the survival rate. Immunotherapy in combination with other forms of cancer therapy presents a formidable arsenal in annihilating cancer, as well as prevents its recurrence and represents the next-generation “smart” cancer therapeutics. However, it is also essential to address scale-up issues and establish the long-term efficacy, bioavailability, and tolerance of payloads through rigorous clinical trials.

## Figures and Tables

**Figure 1 vaccines-11-00458-f001:**
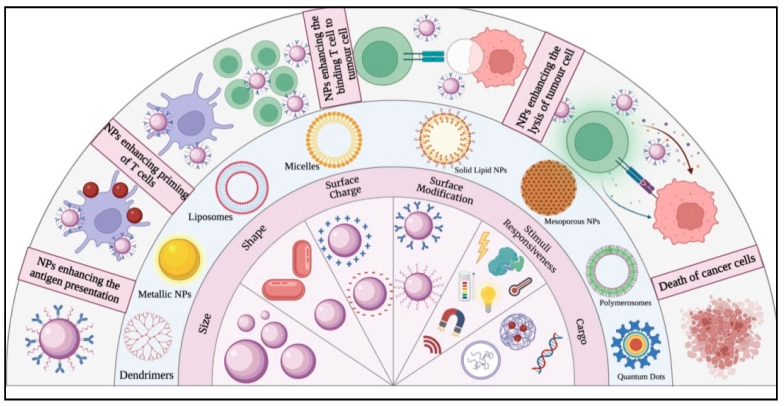
Recently explored nanoparticles for cancer immunotherapy using different strategies and modifications.

**Figure 4 vaccines-11-00458-f004:**
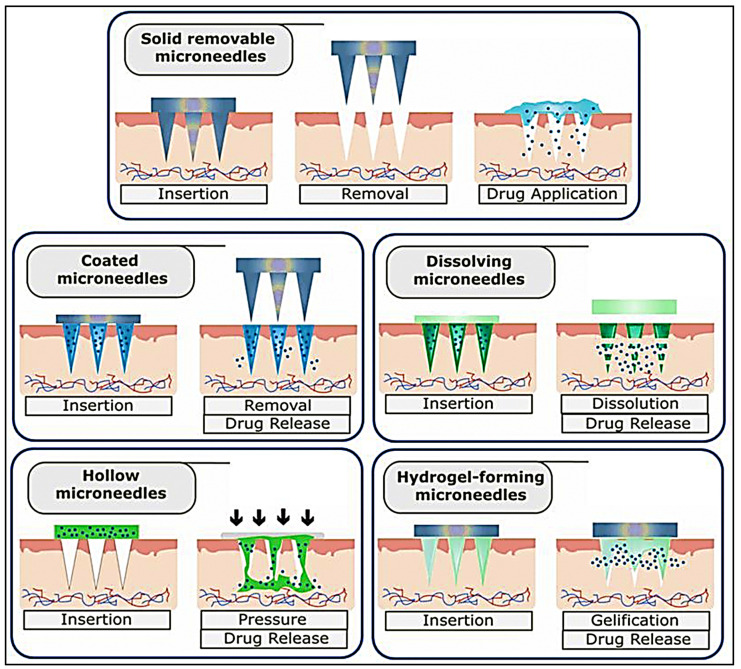
Different types of microneedles employed in the controlled delivery of immunotherapeutic drugs [82]. Reproduced with permissions from Hamed Amani et al., *Journal of controlled release*, Elsevier, 2021.

**Figure 5 vaccines-11-00458-f005:**
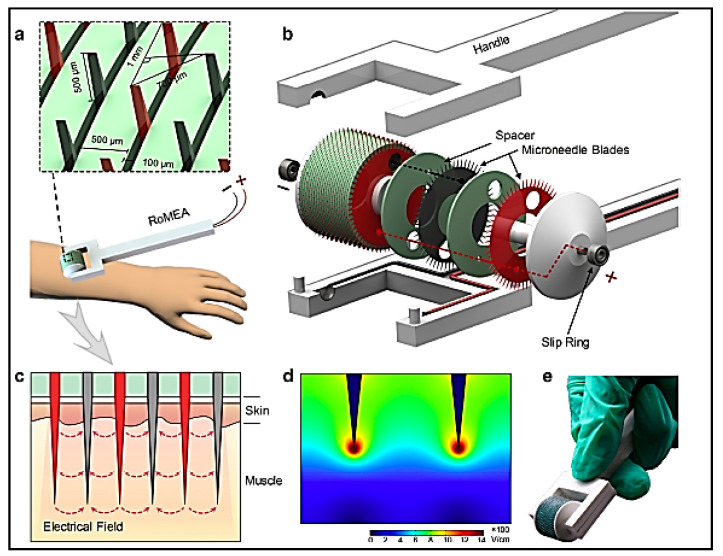
Schematic description of the mechanism of rolling microelectrode array (RoMEA) (**a**) RoMEA’s minimally invasive method of continuous electroporation in a vast region of the target tissue by rolling on the appropriate place. (**b**) Overall design of RoMEA device. Anode (red) and cathode (black) are connected by two nearby microneedle blades. (**c**) Microneedle electroporation. (**d**) Stimulation of RoMEA by electric field (50 V). (**e**) The RoMEA prototype [90]. Reproduced with permissions from Tongren Yeng et al., *Nano Today*, Elsevier, 2021.

**Figure 6 vaccines-11-00458-f006:**
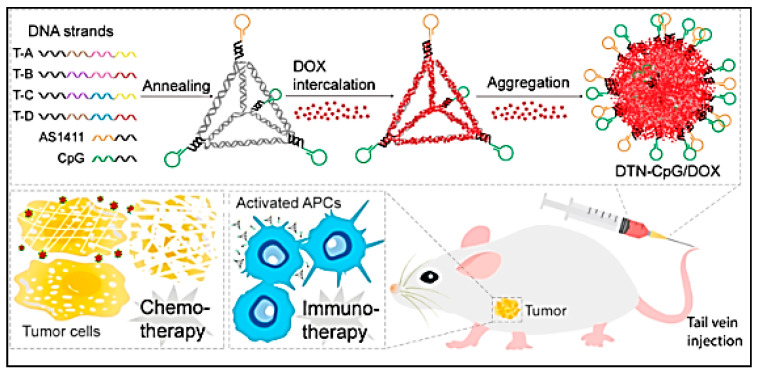
Development of a DNA nanocarrier co-encapsulating CpG and doxorubicin for chemoimmunotherapy. The nanoparticle administration offered superior therapeutic outcome due to the synergistic action of the activated antigen presenting cells and cytotoxicity of the chemotherapeutic agent [96]. Reproduced with permissions from Qian Wang et al., *ACS Applied Nanomaterials*, American Chemical Society, 2022.

**Figure 7 vaccines-11-00458-f007:**
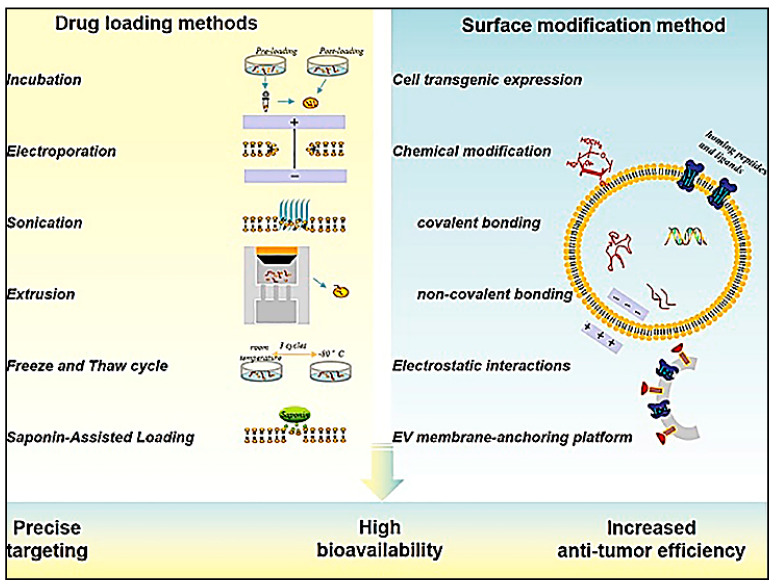
Schematic representation of drug loading of exosomes through the use of culturing, electroporation, sonication, extrusion, freeze-and-thaw cycles, and saponin-assisted loading. Exosome surface modification through chemical alteration, electrostatic interaction, and EV membrane for enhanced targeting efficiency to cancer cells [110]. Reproduced with permissions from Ya-Nan Pi et al., *Biochemical Pharmacology*, Elsevier, 2021.

**Figure 8 vaccines-11-00458-f008:**
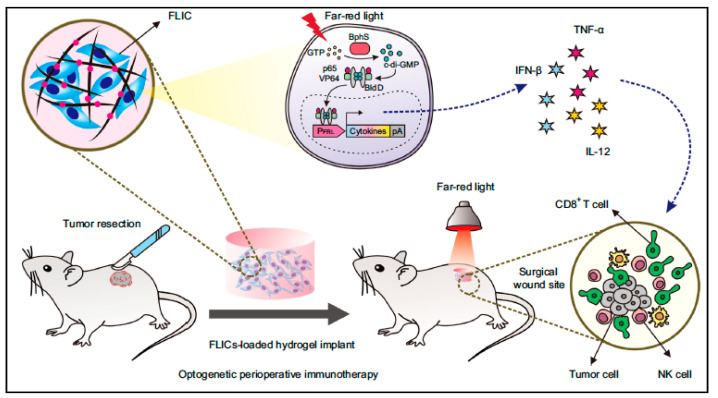
Schematic representation of the in vivo experimental setup for optogenetic perioperative immunotherapy using hydrogel implants loaded with far-red light-controlled immunomodulatory engineered [125]. Copyright CC BY 4.0.

**Figure 9 vaccines-11-00458-f009:**
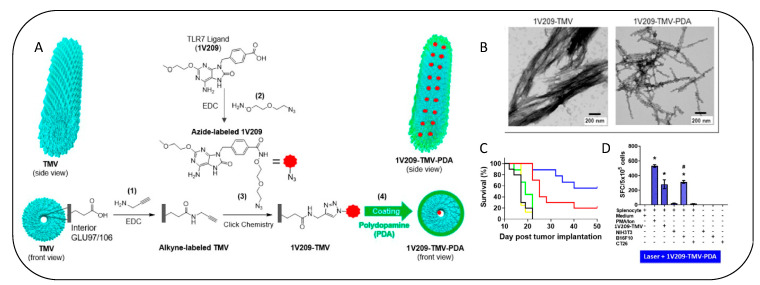
Schematic depiction of the bioconjugation and coating reactions on TMV: (**A**) alkyne labelling of TMV through amidation of the interior glutamate residues of TMV with propargyl amine; (**B**) azide labelling of 1V209 through amidation of the carboxylic group with aminooxy-PEG1-azide; (**C**) 1V209-TMVproduction by copper-mediated azide-alkyne cycloaddition (CuAAC); and (**D**) polydopamine coating of the prepared 1V209 (Tris buffer, pH 8.5) using oxidative polymerization [134]. The results were analyzed with One-way ANOVA followed by Dunnett′s post-test: * *p* < 0.0001 vs. medium; # *p* = 0. Reproduced with permissions from Christian Isalomboto Nkanga et al., *Nanomedicine: Nanotechnology*, *Biology and Medicine*, Elsevier, 2022.

**Figure 10 vaccines-11-00458-f010:**
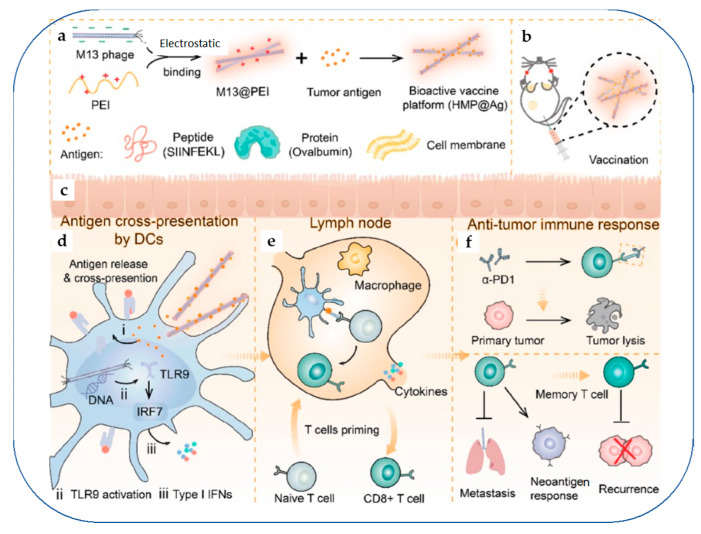
Representation of M13 phage-based vaccine platform design. (**a**) Development of the hybrid M13 phage vaccine (**b**,**c**) loaded with tumour antigens. Schematic diagram representing the stages of antitumour immune response induced by HMP@Ag vaccine. Following subcutaneous administration in mice, DCs internalized the HMP@Ag vaccine for antigen release and cross-presentation for DC maturation (**d**). Mature DCs move to lymph nodes where CD8^+^ T lymphocytes specific for the antigen get activated and expanded (**e**). Combination of M13 phage-based vaccination with ICB therapy inhibits both primary and metastatic cancers and elicit a neoantigen-based CTL response, as well as suppresses tumour recurrence following surgery (**f**) [135]. Reproduced with permissions from Xue Dong et al., *Biomaterials*, Elsevier, 2023.

**Figure 11 vaccines-11-00458-f011:**
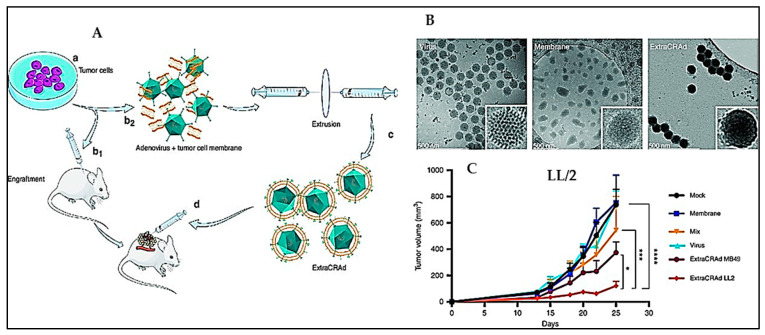
(**A**) Tumour cells (a) were cultured and engrafted into mouse model (b_1_). (b_2_) The cell membrane was extracted and mixed with an oncolytic adenovirus serotype 5, with a 24-base-pairs deletion, carrying -CpG islands (i.e., A5-Δ24-CpG). (c) The virus was wrapped with the cell mem brane using the process of extrusion to obtain ExtraCRAd. (d) The established tumours were treated with multiple intratumoral injections of ExtraCRAd. (**B**) Cryo-transmission electron microscopy (TEM) images of virus, lipid cancer membrane vesicles, and ExtraCRAd (**C**) Median tumour growth [138]. The results were analyzed with a two-way ANOVA, and Dunnet’s post-test comparison, and the levels of significance were * *p* < 0.05, *** *p* < 0.001, **** *p* < 0.0001. Copyright CC BY 4.0.

**Figure 12 vaccines-11-00458-f012:**
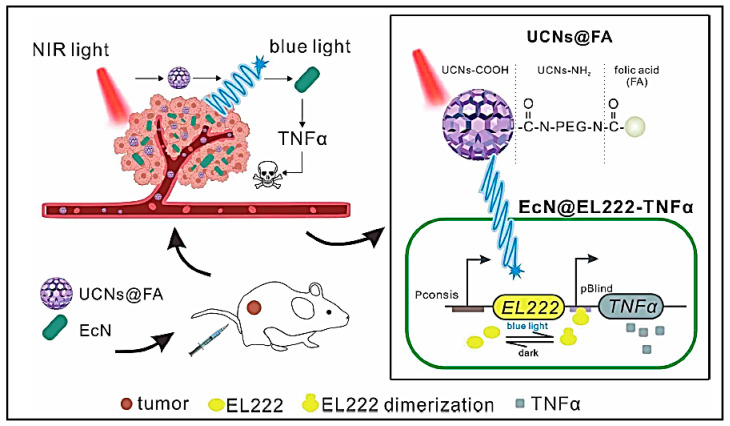
Scheme depicting the upconversion optogenetic system (UCNs@FA and NIR light) and engineered NIR-light-responsive bacteria (EcN@EL222-TNF) [141]. Reproduced with permissions from Huizhuo Pan et al., *Chemical Engineering Journal*, Elsevier, 2021.

**Figure 13 vaccines-11-00458-f013:**
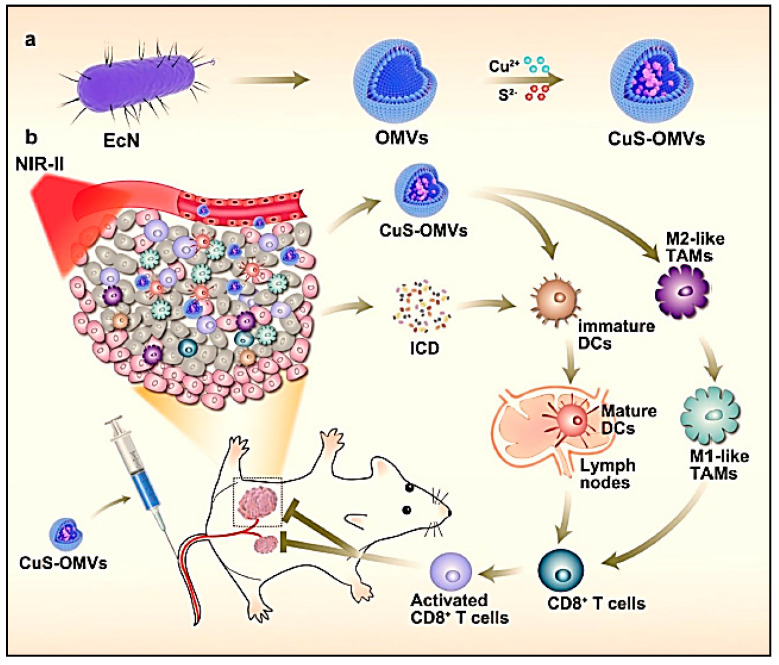
CuS-OMVs for combinational cancer therapy involving PTT and immunotherapy. (**a**) Schematic representation of the preparation of CuS-OMVs. (**b**) Schematic representation of PTT and antitumour immunity generated by CuS-OMVs following NIR–II light exposure. CuS-OMVs target tumours effectively and cause cytotoxicity in tumour cells due to a combination of ICD, DC maturation and CD8^+^ T-cell activation in response to NIR-II light exposure. CuS-OMVs act as immune adjuvants that support DC maturation and repolarize TAMs from M2 to M1 phenotype to inhibit tumour growth and metastasis [146]. Reproduced with permissions from Jiaqi Qin et al., *Nano Today*, Elsevier, 2022.

**Figure 14 vaccines-11-00458-f014:**
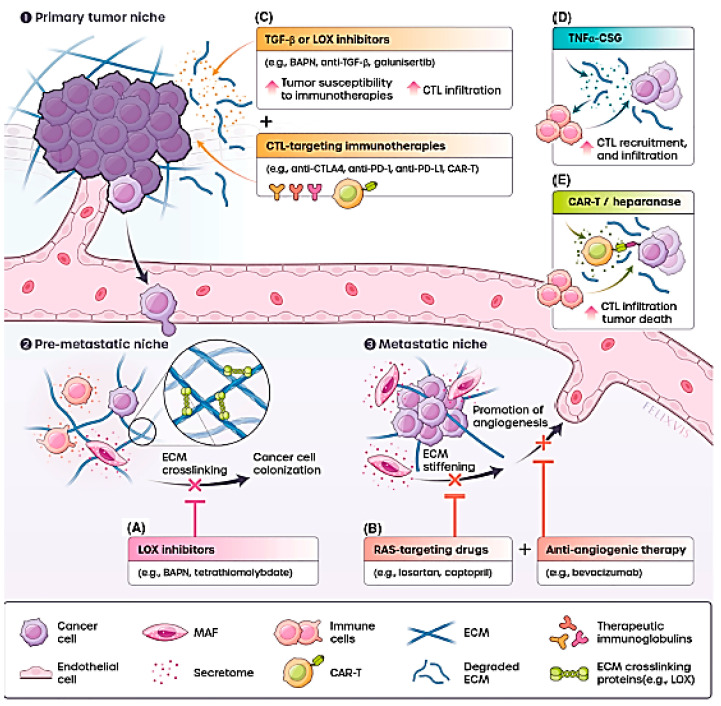
Targeting stiffness in primary and metastatic tumour niches. (**A**) Lysyl oxidase (LOX) inhibitor therapy prevents tumour metastasis by destructing the premetastatic niche, thereby inhibiting cancer cells from colonization. (**B**) Antiangiogenic therapy for the treatment of metastatic cancer by drugs that target the renin-angiotensin system (RAS), and inactivating metastasis-associated fibroblasts (MAFs), which stimulate angiogenesis in the metastatic niche. (**C**) Tissue-softening techniques combined with CTL-targeting immunotherapies, such as immune checkpoint inhibitors and chimeric antigen receptor-T (CAR-T) cells, the infiltration of cytotoxic T lymphocytes (CTLs) increases. (**D**) Recruitment of CTLs into tumour lesions is accompanied by the degradation of tumour ECM by the fusion protein (TNF-CSG), which binds to laminin-nidogen complexes in tumour ECM. (**E**) Heparanase, an enzyme that breaks down the ECM, is expressed by CAR-T cells, which increases their capacity to penetrate the ECM [156]. Reproduced with permissions from Jeongeun Hyun et al., *Trends in Molecular Medicine*, Elsevier, 2022.

**Table 1 vaccines-11-00458-t001:** Nanocarrier mediated immunotherapy interventions and their effect on various cancers.

Type of Therapy	Nanoparticle	Drug/Reactive Component	Cancer Type	Effect	References
Cancer Vaccine	Hydrogel	CaCO_3_	TNBC	DC maturation and T-cell activation	[30]
	OVA-EPC-Span85 complex	OVA	Mouse lymphoma	Activates both cellular and humoral immunity	[31]
	Hydrogel-encapsulated GM-CSF, CpG-ODN	GM-CSF, CpG-ODN, a TLR 9 agonist, and tumour cell lysates	Mouse colon carcinoma and Melanoma	Dendritic cell maturation and Immune system activation	[32]
	CaP-peptide vaccine	Calcium phosphate (CaP) and Peptides	Colon cancer and Breast cancer	Dendritic cell maturation	[20]
	CS/γ-PGA nanoparticle	MUC1 glycopeptide antigens	Breast cancer	Produce significantly high titers of IgG antibody	[33]
	A novel polyethyleneimine (PEI)-based personalized vaccine—NP vaccination combined with STING agonist therapy	Neoantigen peptides and CpG adjuvants in a compact nanoparticle	Colon carcinoma and melanoma	Tumour infiltration of CD8^+^ T cells	[27]
	CTX-loaded hydrogel and PLEL hydrogel	CpG and tumour lysates	Colon carcinoma	Produces the cytotoxic T lymphocyte and Immunogenic cell death	[34]
	Fe_3_O_4_ nanocomposite	OVA	Melanoma	Efficiently stimulate dendritic cell-based immunotherapy and potentially-activate macrophages	[35]
	CaCO_3_ Nanoparticle	CaCO_3_@(OVA/HPAA-CpG)3 vaccines	Lymphoma	Dendritic cell maturation and CD8^+^ T-cell proliferation	[36]
	A PEG derivative (PpASE) stabilized aluminium nanoparticle for delivering the synthetic long peptides (ANLs)	ANLs ANSs	Melanoma	Activation and proliferation of CD8^+^ T cells	[37]
	Mn-NP (Carrier and adjuvant)	OVA (Model Antigen), CpG (Adjuvant), Anti-PDL1	Melanoma	Activation of the cGAS-STING pathway. Nanovaccine (NV) or Personalized NV (s.c.) Anti-PD-L1 (i.v.)	[38]
	DGBA-OVA-CpG nanovaccine	unmethylated cytosine-guanine dinucleotides (CpG) (adjuvant)	Melanoma	Controlled tumour growth along with anti-PD1 checkpoint inhibition	[39]
	Bi-specific macrophage nano-engager (BiME)	Serum albumin and targeted moiety	Melanoma	Robust T-cell activation	[40]
	Nanotransformer- based vaccine with anti-PD-L1 antibodies	A polymer–peptide conjugate-based nanotransformer and loaded antigenic pep	Melanoma	Activates the NLRP3- inflammasome pathway and thus boosts antitumour immunity and stimulation of CD8^+^ T cells	[41]
Immunotherapy	Tumour exosomes (TEX)	HSP70, HSP90, MHC I, MHC II, TGF-β, and PD-L1	TNBC	Dendritic cell activation, Cytotoxic T-cell-mediated immune response	[24]
	Magnetic nanocomplexes (Iron oxide)	-	TNBC	STING activation and Macrophage polarization	[42]
	Folic acid conjugated superparamagnetic iron oxide, Trimethyl chitosan (TMC) nanoparticles	EZH2/CD73 siRNA	TNBC	Gene silencing	[43]
	LPS-decorated PLGA nanoparticles	LPS	Murine colon adeno-carcinoma and glioma	Activation of TLR4 Macrophage and DCs Proliferation	[44]
	MUC1-Dex	-	Melanoma	Activation of CD8^+^ T cells	[45]
	ZNPs/I@CML	Indomethacin	Prostate cancer	ZSTK is an effective pan-PI3K inhibitor, Macrophage polarization	[46]
	Cargo-free PLG nanoparticles	Anti-PD-L1 antibody	TNBC	Decrease the expression of MCP-1 by 5-fold and increase the expression of TNF-α by more than 2-fold upon uptake by innate immune cells	[47]
	Poly (beta-amino ester) (PBAE) nanoparticle	Cyclic dinucleotides (CDNs)	Melanoma	Stimulator of interferon receptor (STING) enhanced cancer immunotherapy	[48]
	UPP@OVA complex	Yb and Er-doped NaY/GdF4 UCNPs	Melanoma	Enhanced T-cell proliferation, interferon gamma production and cytotoxic T lymphocyte (CTL) mediated responses	[49]
	Split bullet nanoparticle	Doxorubicin and iRGD peptide	Melanoma	Suppress primary melanoma and initiate immune memory against tumour recurrence	[50]
	pH sensitive liposomes	Pyranine and antigenic protein Ovalbumin (OVA)	Lymphoma	Increased specific immunity and tumour regression occurred	[51]
Immune checkpoint inhibitor (ICI) therapy	Z-domain conjugated ferumoxytol nanocarrier	Nanointerface (aPD-L1-Z-Fer)	Hepato-cellular carcinoma	Block the PD-1/PD-L1 (Programmed death ligand)	[52]
Immunogene therapy	Miktoarm star polymer (PDMAEMA-POEGMA) nanoparticles	βIII-tubulin, Polo-Like Kinase 1 (PLK1)—siRNA	NSCLC	Gene silencing	[53]
	Methoxypoly (ethylene glycol)—Poly(caprolactone) was hybridized with Dimethyldioctadecyl-ammonium bromide (DDAB) cationic lipid (mPEG-PCL-DDAB) nanoparticles“mPEG-PCL-DDAB nanoparticle”	Anti-insulin-like growth factor 1 receptor-siRNA and lycopene	Breast cancer	Apoptosis and arrested cell cycle	[54]
Chemoimmunotherapy	Pep-PAPM	Anti-PD-L1 peptide and Paclitaxel	TNBC	PD-L1 blockade and ROS-induced damage	[55]
	231MARS@PLGA	PD-L1 inhibitor and Paclitaxel	TNBC	Affect the tumour stiffness	[56]
	SK/siTGF-β NPs	Shikonin and siTGF-β	TNBC	Dendritic cell activation, Cytotoxic cell-mediated immune response	[57]
	PEG-b-PNHS polymer-conjugated 5-ASA (PASA) Folate-PEG-NH_2_-conjugated PASA (FASA)	5-ASA and DOX	Mouse breast and colon cancer models	Anti-PD-L1 Activation. Macrophage activation and proliferation	[58]
	Ferritin nanocages	PD-L1pep1 and Doxorubicin	Human breast tumour and mouse colon tumour	Inhibited PD-1/PD-L1 interaction and restored T-cell activity	[59]
	Nano assembly	JQ1/Rapa-IR783	TNBC	Co-inhibition of PD-L1/mTOR	[60]
	Doxorubicin/CpG self-assembled nanoparticles	Doxorubicin/CpG self-assembled nanoparticles, prodrug and dendritic cells (DC) co-encapsulated hydrogel system	Melanoma	Enhanced antigen presentation in DCs and CTL mediated tumour killing	[28]
	Nano-Folox (Nanoprecipitate of Folinic acid and Oxaliplatin)	Folinic acid (FnA), 5-fluorouracil (5-Fu), and oxaliplatin (OxP)	Colorectal cancer and hepatocellular carcinoma	Induce apoptosis and immunogenic cell death	[61]
	Nano-emulsion	Puerarin (nanoPue) and paclitaxel	TNBC	Deactivated tumour-associated fibroblast (TAFs) and 2-fold times increased the intra-tumoural infiltration of cytotoxic T cells	[62]
Chemotherapy and immune checkpoint blockade therapy	BMS/RA@CC-Liposome	Chemotherapeutic drug (RA-V) and PD-1/PD-L1 blockade inhibitor (BMS-202)	Colorectal carcinoma	Dendritic cell maturation, Cytotoxic T-cell-mediated immune response	[63]
	A metabolism nano-intervenor of DCs (Man-OVA(RSV) NPs) was loaded in a versatile hydrogel system	Metformin hydrochloride (MET), Rosuvastatin (RSV)	Melanoma	DC-mediated immunotherapy	[26]
Exocytosis blockade of ER along with anti-PD-L1 therapy	Homologous cancer cell membrane coated nanoparticle (HCC@NP)	Brefeldin A (BFA)	Melanoma	Antitumour immunity and reversing immune suppression	[64]
Radioimmunotherapy	Hybrid nanoplatform (MGTe) composed of gTe (glutathione (GSH) decorated Te nanoparticles)	gTe was designed for radiotherapy sensitization, concurrently the fusion of TM and BM was expected for amplifying antitumour immune response	Breastcancer	X-Ray irradiation: ROS production and Immunogenic death (ICD) APC maturation and T-cell stimulation.	[65]
	Chitosan/γ-PGA nanoparticles	-	TNBC	Decrease in the percentage of immunosuppressive myeloid cells and an increase in the antitumoural CD4^+^IFN-γ^+^ population	[66]
Photothermal immunotherapy	Nano modulator IQS (ICG/JQ1/BMS nanoparticles)	ICG/JQ1/BMS	Mouse colon carcinoma	Immunogenic cell death (ICD) upon laser irradiation (PTT) and dual-block PD-L1 and IDO-1 pathways	[67]
	Prussian blue nanoparticles (PBNP)	CpG-PBNP-PTT	Neuro-blastoma	T-cell activation and robust memory generation	[68]
	Polydopamine–Mesoporous Silica Core–Shell Nanoparticles	Polydopamine nanoparticle—Photothermal agent Gardiquimod—Immunomodulatory drug	Murine melanoma	Photothermal ablation of the cancer cells	[69]
	ICG-loaded magnetic nanoparticles (MIRDs)	Polyethylene glycol polyphenols (DPA-PEG)-R837 loaded	Breast cancer	Inhibited tumour growth and metastasis and recurrence	[70]
Photodynamic Immuno therapy	Nano-booster (NC@Ce6)	Anti-programmed death-ligand 1 (aPDL1) and photosensitizer (Ce6) into the acid-responsive nanocomplex (NC)	Melanoma	ROS generation and Immunogenic cell death. Increases the intra-tumoural infiltration of CD8^+^ T cells	[22]
	PyroR	Photosensitizer pyropheophorbide-a (Pyro) and TLR agonist resiquimod (R848)	Breast cancer	ROS generation. Dendritic cells (DCs) maturation and activate cytotoxic T lymphocytes. R848 induces macrophage repolarization.	[23]
	Hybrid CTTPA-G using cancer cell membranes (CC-Ms) and mesoporous silica nanoparticles (MSNs)	Type I AIE photosensitizer (TTPA) and glutamine antagonist	Melanoma	Regulate nutrition partitioning and remodelling the immune suppressive microenvironment	[71]
Ferrotherapy and immunotherapy	Nanoparticle—fusion of hepcidin and leukemia cell membrane vesicles on gold nanoparticles (AuNPs)	Hollow mesoporous Prussian blue (AuPB@LMHep)	Leukemia	Immune response amplification via Ferrotherapy against tumour	[72]
Chemophotothermal therapy	Hollow gold nanostars (HGNSs) and gold nanocages (GNCs)	Doxorubicin	Breast cancer	Apoptosis	[73]
Photoimmunotherapy (Photodynamic/photo-thermal and immune-modulatory effects)	Nanoporphyrin platform	Mouse mAb anti-PD-L1	TNBC	Sensitizing the “cold” tumour microenvironment via laser therapy followed by Immune checkpoint Blockade (PD-L1 blockade)	[74]
	Black phosphorus and PEGylated Hyaluronic acid (HA-BP nanoparticle)	HA-BP	TNBC	Macrophage polarization. Immunogenic cell death and maturation of DCs	[24]

**Table 2 vaccines-11-00458-t002:** Emerging strategies in cancer immunotherapy and their salient features.

S. No.	Therapy	Nanoparticle/Material	Salient Characteristics	Cancer Model	Reference
1	Microneedle based Immuno-therapy	Rolling stainless steel microneedle electrode array (RoMEA)	Merit: Efficient siRNA delivery and gene silencing. Demerit: Currently, RoMEA is limited to nucleic acid delivery only	B16F10/CT26 xenograft mouse models	[90]
		Hyaluronic acid integrated with pH-sensitive dextran nanoparticles (NPs) encapsulating anti-PD1 and glucose oxidase (GOx)	Merit: Triggered release of anti-PD1 antibody and immunomodulators (anti-CTLA4). Demerit: Focused only on skin cancer	B16F10 mouse model	[87]
		pH-responsive tumour-targeted lipid nanoparticles (NPs)	Merit: Local delivery of aPD-1 and cisplatin Demerit: Shelf-life and stability issues	SCC VII mouse model	[86]
		F127 nanoparticles loaded with R837 and coated with cancer cell membranes	Merit: Suppression of tumour growth by inhibiting angiogenesis	HCT116 mouse model	[83]
		DNA vaccine delivery system with a layer-by-layer coating of ultra-pH-responsive OSM-(PEG-PAEU) and immunostimulatory adjuvant	Merit: Increase the immunogenecity Demerit: Risk of affecting host genome	B16/OVA melanoma tumours in mouse model	[85]
2	Nucleic acid-mediated therapy	Nucleic acid nanoassembly (NAN)-based technology for functionalization of hydrogels using isothermal toehold-mediated reassociation of RNA/DNA heteroduplexes.	Merit: Efficient capture of human T-lymphocytes and tunable activation of TCR Demerit: No in vivo studies for validation	-	[166]
		Immunostimulatory DNA hydrogel consisting of a hexapod-like structured DNA (hexapodna) with CpG sequence and gold nanoparticles	Merit: Interferon- gamma production from splenocytes. Demerit: Irradiation causes adverse effects	EG7-OVA tumour-bearing mouse model	[92]
		Targeted nano vaccine equipping cell membrane vesicles (CMVs) from tumour cells with functional DNA, including CpG oligonucleotide	Merit: Long-term immune memory to prevent tumour recurrence Demerit: Isolation of CMVs is dificult due to tumour heterogeneity	B16-OVA tumour-bearing mice	[94]
		pH-driven interlocked DNA nano-spring (iDNS) to stimulate T-cell activation	Merit: Spring-like shrinking of iDNS leading to antitumour effect Demerit: Challenge to merge functional DNA building blocks	B16F10 tumour-bearing mice	[95]
		DNA tetrahedron to create a nanoplatform for co-delivery of drug doxorubicin and the CpG oligodeoxy-nucleotides	Merit: Synergistic therapeutic effects and pronounced antitumour efficiency Demerit: DNA tetrahedron might not be able to carry long nucleic acids	B16F10 tumour-bearing mouse model	[96]
3	Gene editing	Programmable unlocking nano-matryoshka-CRISPR system (PUN) targeting programmed cell death ligand 1 (PD-L1) and protein tyrosine phosphatase N2 (PTPN2)	Merit: PUN exhibits optimal antitumour efficiency and long-term immune memory Demerit: Xenograft tumour model used	B16 tumour-bearing mouse model	[104]
		Nanoassembled ribonucleoprotein complexes (NanoRNP), which can efficiently block the PD-L1 immune checkpoint	Merit: Sustained downregulation of PD-L1	B16F10 tumour-bearing mouse model	[102]
		Lipid nanoparticle complexed to plasmid DNA co-encoding CRISPR-associated protein 9 and LDHA-specific sgRNA, to form the lipoplex, pCas9-sgLDHA/F3	Merit: Treatment activated the interferon-gamma and granzyme production of T cells in culture Demerit: Transfection mechanism of F3 not known	B16F10 tumour-bearing mouse model	[108]
		Specific promoter-driven CRISPR/Cas9 system, F-PC/pHCP, achieves permanent genomic disruption of PD-L1	Merit: Disrupts the PD-L1 gene preventing immune escape	B16F10 tumour-bearing mouse model	[105]
		HPT-PFs modified with hyaluronic acid (HA) and tumour microenvironment sensitive peptides (TMSP)	Combined CD47 knockout with IL-12 production, leads to significant inhibition of tumour growth	B16F10 tumour-bearing mouse model	[107]
4	Exosomes	Surface-engineered antigenic exosomes using melanoma tumour peptides	Merit: Induced antigen- specific CD8 T cell proliferation	2 Pmel 1 transgenic mice	[112]
5	Engineered cells	Paclitaxel-loaded fake blood cell Eudragit particle (Eu-FBCP/PTX)	Merit: Exhibits better phagocytic and micropinocytic uptake	MC-38 tumour models	[25]
		Bone marrow-derived mesenchymal stem cells (MSCs) engineered to express the immune stimulating factor LIGHT	Merits: LIGHT- expressing MSCs exhibit potent antitumour immune response; Reverses immunesuppressive TME	TUBO (murine mammary carcinoma)	[117]
		Dibenzocyclooctyne- poly(ethylene glycol)- pheophorbide A conjugated to human mesenchymal stem cell (hMSC-DPP)	Merits: hMSC-DPP recognizes cancer lesions, mediates cell death by irradiation; Immune regulation at the target site	K1735 tumour-bearing mouse model	[116]
6	CAR-T Therapy	Stem cells engineered to stably express various chimeric antigen receptors (CARs) against tumour-associated antigens	Merits: Long-term immune cell generation, sustained tumour-specific effector cells to maintain remission Demerit: T cells in human thymus may not cause immune-tolerance to the mouse host	BLT (Bone, Liver, Thymus) humanized mouse model	[115]
7	Nano-optogenetics	Pan-T cells (Human Peripheral Blood CD3+ T Cells) transduced using pCDH-OPN4-eGFP and pNFAT-3CK construct exposed to blue light illumination	Merits: Photo- activatable engineered T cells suppressing tumour growth; Cytotoxicity increases with blue light illumination Demerit: Low penetration depth of blue light	Subcutaneous xenografts in hepatocellular carcinoma	[124]
		Far-red light-controlled immunomodulatory engineered cells (FLICs) loaded into a hydrogel scaffold	Merits: FLICs-loaded hydrogel implants elicit long-term immunological memory; Prevents tumour recurrence Demerit: Determination of T cell response was only carried out ex vivo; In vivo response not known	B16F10 ovalbumin expressing melanoma model	[125]
		Engineered bacteria EcN-pDawn-φx174E/TRAIL	Merits: Both diagnosis and light—controlled cancer therapy Demerit: Poor penetration depth of blue radiation	Colorectal cancer theranostic and therapy	[142]
8	Virus and viral components for Immunotherapy	Papaya mosaic virus nanoparticle (PapMV)	Merits: Synergistically improves the therapeutic effect; PapMV alone induced the development of CD8^+^ T-cell responses against endogenous tumour epitopes Demerit: Intratumoral imjection for antitumour activity performed and may not be applicable to deep-seated tumours	Subcutaneous (B16-OVA) melanoma model	[130]
		Cowpea mosaic virus (CPMV)	Merits: In situ vaccine modulates the TME potentiate antitumour immunity; Exhibits excellent antitumour activity when compared to other visruses Demerit: Antitumour response depends on the capsid viral protein recognition	Colon cancer, Melanoma, Ovarian cancer model.	[132]
		Tobacco mosaic virus (TMV) conjugated with toll-like receptor 7 agonist (1V209), and surface- coated with photothermal biopolymer polydopamine (PDA)	Merits: Intratumorally injected and irradiated using an 808 nm near-infrared laser enhances antitumour activity; Inhibition of tumour recurrence Demerit: Long-term effects due to irradiation not known	B16F10 dermal melanoma mice	[134]
		SeV (sendai virus) + aCD47)@PLGA nanoparticles	Merits: Nano-composite strategy enhances antitumour efficacy by TME; Immuno- modulation suppresses tumour metastasis and recurrence Demerit: Intratumoral injection performed; May not be applicable to deep seated tumours	4T1 murine mouse model	[129]
9	Oncolytic virus based immunotherapy	Virus artificially wrapped with tumour cancer membranes carrying tumour antigens	Merits: Increased infectivity and oncolytic effect; controls the growth of aggressive melanoma and lung tumours	Subcutaneous murine model of melanoma and lung cancer	[138]
10	Bacterial Immuno- therapeutics	Hybrid vaccine platform (HMP@Ag) using hybrid M13 phage and personal tumour antigens	Merits: Activation of antigen-presenting cells (APCs) through the Toll-like receptor 9 (TLR9) signaling pathway; Uses personalised antigens Demerit: Pathogenicity of bacteria might induce immune-related adverse events	B16-OVA melanoma model	[135]
		Outer membrane vesicles (OMVs) from *Escherichia colbiomimetic* containing copper sulfide nanoparticles are fabricated (CuS-OMVs).	Merits: Induced strong immunogenic cell death (ICD) of tumour cells; Acts as immune adjuvant and causes repolarisation of TAMs Demerit: Long-term safety of CuS not known	Murine 4T1 breast cancer model	[146]
		Gold nanoparticles (AuNPs) cloaked within the outer-membrane vesicles (OMVs) from *E. coli*	Merits: Cytotoxic effect on GL261 glioma cells; low—dose combination radiotherapy Demerit: Need to explore the affinity of AuNPs and OMVs	Subcutaneous tumour model and In situ brain tumour model	[145]
		Engineered biotherapeutic platform using EcN-EL222-TNFα and UCNs@FA	Merits: Upconversion optogenetic strategy enhances anti—tumour efficacy; Controllability and biocompatibility, deeper penetrability Demerit: Long-term effects of therapy unknown	4T1—tumour-bearing mouse model	[141]
11	Fungal based Immunotherapeutic	Fungal beta-glucans	Merit: Stimulates both innate and adaptive immune responses Merit:.Direct cytotoxic effect Demerits: Toxicity studies are not performed; Orthotopic models not studied	Xenograft colon cancer	[148]
		Nanoparticle incorporated polysaccharide mannan structure from *Saccharomyces cerevisiae*	Merit: Strongly induces T helper 17 (T_H_17) responseDemerit: Intratumoral injection carried out; May not be applicable to deep seated tumours	Colon and melanoma cancer	[149]
		Immuno modualting polysaccharides from *Inonotus obliquus*	Merit: Transforms TAMs into proinflammatory phenotype Demerit: Systemic characterisation of immunological properties is lacking	*In vitro* studies by co-culture of using mouse lung cancer cell lines and macrophages	[147]
12	Herbal interventions for Immunotherapy	Enzyme-sensitive tumour-targeting nano drug delivery system AP-PP-DOX (Polysaccharides from *Angelica sinensis* (AP))	Merit: Restores Th1/Th2 immune balance in tumour microenvironment. Demerit: In vivo study has not been reported	-	[167]
		Innate immune activator Astragaloside III (As) photosensitizer chlorine e6 (Ce6) ((As + Ce6)@MSNs-PEG)	Merits: Effectively activates NK cells and inhibits the proliferation of tumour cells in vitro; Induces infiltration of immune cells into the tumour; Enhances the cytotoxicity of natural killer cells and CD8+ T cells in vivo	CT26 tumour-bearing model	[152]
13	3D matrix architecture for Immunotherapy	Three-dimensional (3D) poly(ethylene)glycol (PEG) hydrogels covalently combined with low molecular weight heparin	Merits: PEG provides structural and mechanical property, anchoring of CCL21 to heparin influences cell migration and proliferation; T cells reproduce in large number Demerit: In vivo study has not been reported	-	[163]

**Table 3 vaccines-11-00458-t003:** Clinical trials (completed and ongoing) involving nanoparticles for cancer immunotherapy.

S. No	Clinical Trial	Cancer Type	Therapeutic Intervention	Clinical Trial ID
1	A study to evaluate/tolerability of immune-therapy combinations in participation with TNBC or gynaecologic malignancies (Completed)	TNBC and Ovarian cancer	Etrumadenant (antagonist of immunomodulatory checkpoint molecules adenosine A2A and A2B receptors), Eganelisib (IPI-549, phosphoinositide 3-kinase inhibitor), PEGylated liposomal doxorubicin (PLD), albumin nanoparticle-bound paclitaxel (NP)	NCT03719326
2	Neoadjuvant LDRT combined with Durvalumab in potentially resectable stage III NSCLC (Ongoing)	Stage III NSCLC	Durvalumab (immune checkpoint inhibitor antibody), Albumin nanoparticle-bound paclitaxel along with low-dose radiation therapy	NCT05157542
3	Dose escalation study of immunomodulatory nanoparticles (Ongoing)	Advanced solid tumours	PRECIOUS-01 (invariant natural killer T cell activator threitolceramide-6 and New York Esophageal Squamous Cell Carcinoma-1 cancer-testis antigen peptides encapsulated in PLGA nanoparticle)	NCT04751786
4	A pilot study of neoadjuvant chemotherapy with or without Camrelizumab for locally advanced gastric cancer (Ongoing)	Gastric cancer	Camrelizumab (anti-PD1) and chemotherapy with albumin nanoparticle-bound paclitaxel and oxaliplatin	NCT05101616
5	NBTXR3, Radiation Therapy, and Pembrolizumab for the treatment of recurrent or metastatic head and neck squamous cell cancer (Ongoing)	Metastatic head and neck squamous cell carcinoma and recurrent head and neck squamous cell carcinoma	Hafnium oxide containing nano- particles (NBTXR3) with hypo-fractionated radiation therapy and Pembrolizumab (anti-PD1 humanized antibody) with Stereotactic body radiation therapy	NCT04862455
6	Radiation therapy to the usual treatment (Immunotherapy with or without chemotherapy) for Stage IV non-small cell lung cancer patients who are PD-L1 negative (Ongoing)	Advanced lung adenocarcinoma, Advanced lung adenosquamous carcinoma, Advanced and metastatic lung NSCC (Stages IIIB/IIIC/IV/IVA/IVB), Metastatic lung adeno-carcinoma, Metastatic lung adeno-squamous carcinoma, lung cancer AJCC v8	Carboplatin, Ipilimumab (CTLA4 targeting antibody), albumin nanoparticle-bound paclitaxel, Nivolumab (anti-PD1), Paclitaxel, Pembrolizumab, Pemetrexed along with Stereotactic body radiation therapy	NCT04929041
7	Gemcitabine, Nab-paclitaxel, Durvalumab, and Oleclumab before surgery for the treatment of in resectable/borderline resectable primary pancreatic cancer (Ongoing)	Borderline resectable pancreatic adeno-carcinoma, Resectable pancreatic adeno-carcinoma (IA/IB/IIA/IIB) pancreatic cancer AJCC v8	Durvalumab, Gemcitabine, albumin nanoparticle-bound paclitaxel, Oleclumab (anti-CD73)	NCT04940286
8	Combination with chemotherapy for the treatment of advanced solid tumours involving the abdomen or thorax (Ongoing)	Advanced breast carcinoma, Advanced endometrial carcinoma, Advanced fallopian tube carcinoma, Advanced hepatocellular carcinoma, Advanced malignant abdominal neoplasm, Advanced malignant female reproductive system neoplasm, Advanced malignant thoracic neoplasm, Advanced ovarian carcinoma, Advanced primary peritoneal carcinoma, Advanced renal cell carcinoma	Atezolizumab (anti-PD1), Cabozantinib S-malate (tyrosine kinase inhibitor), albumin nanoparticle-bound paclitaxel	NCT05092373
9	Durvalumab in combination with chemotherapy in treating patients with advanced solid tumours (Ongoing)	Locally advanced malignant solid neoplasm, Metastatic malignant solid neoplasm, Unresectable malignant solid neoplasm	Capecitabine, Carboplatin, Durvalumab, Gemcitabine hydrochloride, Paclitaxel, albumin nanoparticle-bound paclitaxel, PEGylated liposomal doxorubicin hydrochloride	NCT03907475
10	Addition of anticancer drug, ZEN003694 (ZEN-3694) and PD-1 inhibitor (Pembrolizumab) to standard chemo-therapy (Nab-Paclitaxel) treatment in patients with advanced Triple-Negative Breast Cancer (TNBC) (Ongoing)	Anatomic stage III/IV breast cancer AJCC, Locally advanced TNBC, Metastatic TNBC, Unresectable TNBC	BET Bromodomain inhibitor ZEN-3694, albumin nanoparticle-bound paclitaxel, Pembrolizumab	NCT05422794
11	Pembro + Chemo in brain mets (Ongoing)	Lung cancer, Lung cancer metastatic, Brain cancer, Cancer	Pembrolizumab, Paclitaxel, Pemetrexed, Carboplatin, albumin nanoparticle-bound paclitaxel	NCT04964960
12	Atezolizumab with chemotherapy in treating patients with anaplastic or poorly differentiated thyroid cancer (Ongoing)	Metastatic thyroid gland carcinoma, Poorly differentiated thyroid gland carcinoma, Stage IVA/IVB/IVC thyroid gland anaplastic carcinoma AJCC v8, Gland anaplastic carcinoma, Unresectable thyroid gland carcinoma	Atezolizumab (anti-PDL1), Bevacizumab (anti-VEGFA), Cobimetinib (MEK inhibitor), Paclitaxel, Vemurafenib (B-Raf inhibitor), albumin nanoparticle-bound paclitaxel	NCT03181100
13	Local consolidative therapy and Durvalumab for oligoprogressive and polyprogressive stage III NSCLC after chemoradiation and anti-PD-L1 therapy (Ongoing)	Stage III/IIIA/IIIB lung cancer AJCC v8 Stage III/IIIA/IIIB lung Non-Small Cell Cancer AJCC v7	Carboplatin, Durvalumab, Gemcitabine, Paclitaxel, Pemetrexed, albumin nanoparticle-bound paclitaxel	NCT04892953
14	NBTXR3, Radiation therapy, Ipilimumab, and Nivolumab for the treatment of lung and/or liver metastases from solid malignancy (Ongoing)	Advanced malignant solid neoplasm, Meta- static malignant neo- plasm in the liver, Metastatic malignant neoplasm in the lung, Metastatic malignant solid neoplasm	Hafnium oxide-containing nanoparticles (NBTXR3), Ipilimumab (anti-CTLA4), Nivolumab along with radiation therapy	NCT05039632
15	Durvalumab and Tremelimumab in combination with propranolol and chemotherapy for treatment of advanced hepato-pancreabiliary tumours (Ongoing)	Pancreatic Cancer, Hepatocellular Cancer, Biliary Tract Cancer, Cholangiocarcinoma	Durvalumab, Gemcitabine, Tremelimumab, Propranolol, Cisplatin, albumin nanoparticle-bound paclitaxel	NCT05451043
16	Novel RNA-nanoparticle vaccine for the treatment of early melanoma recurrence following adjuvant anti-PD-1 antibody therapy (Ongoing)	Melanoma	Autologous total tumour mRNA loaded DOTAP liposome vaccine	NCT05264974
17	CAR-T cell therapy in relapsed/refractory myeloma with extramedullary disease—an in vivo imaging and molecular monitoring study (CARAMEL) (Ongoing)	Extramedullary Myeloma	JNJ-68284528 (Cilta-cel)—B cell maturation antigen (BCMA) and ^64^Cu SPION dual PET-MR imaging agent	NCT05666700
18	Split course adaptive radiation therapy with Pembrolizumab with/without chemotherapy for treating stage IV lung cancer (Ongoing)	Lung Non-Small Cell Carcinoma, Stage IV lung cancer AJCC v8	Carboplatin, Fludeoxyglucose F-18, Pembrolizumab, Pemetrexed, albumin nanoparticle-bound paclitaxel along with radiation therapy, [18-F] (fluoropropyl)-L-glutamate (FSPG) PET scan	NCT05501665

## Data Availability

Not applicable.

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
