# Peer review of "Emerging Trends in Nano-Driven Immunotherapy for Treatment of Cancer"

_vaccines, 2023, doi:10.3390/vaccines11020458_

Round 1
Reviewer 1 Report
The authors summarized single or combined immunotherapy for cancers by utilizing nanomaterials. The paper is well-summarized and cite enough references. I have small comments.
1. In Table 1, there are many kinds of therapy including single therapy and combination therapy. The appearance order is not well-arranged. It is better to arrange from single immunotherapy to combination therapy. Also, if you explain some contents in manuscript, please insert "Table 1".
2. In latter half contents, the authors describe recent developments in cancer immunotherapy. However, the authors mention "we discuss the recent advances towards nanoparticle-mediated immune engaging systems" in introduction section. The contribution of nanoparticles to other technology like microneedle, cell-based therapy, and virus-mediated therapy, should be summarized. The authors explain the detail in manuscript, but it is better to overview the role of nanoparticles and make table.
3. The citation number should be checked all. The reference number is reverse, for example page 5 ((14)(13), (16)(13)).
4. Some abbreviation appear without spell out, for example page 4 (5 DNA), page 8 (HA). Also abbreviation should be spell out at first appearance.
Author Response
Response to Reviewer-1 comments
The authors summarized single or combined immunotherapy for cancers by utilizing nanomaterials. The paper is well-summarized and cites enough references. I have small comments.
- In Table 1, there are many kinds of therapy including single therapy and combination therapy. The appearance order is not well-arranged. It is better to arrange from single immunotherapy to combination therapy. Also, if you explain some contents in manuscript, please insert "Table 1".
Response: We thank the reviewer for the suggestion. Table 1 is rearranged and organized as suggested in the revised manuscript. Table 1 has been referred in appropriate places in the revised manuscript as suggested.
- In the latter half contents, the authors describe recent developments in cancer immunotherapy. However, the authors mention "we discuss the recent advances towards nanoparticle-mediated immune engaging systems" in introduction section. The contribution of nanoparticles to other technology like microneedle, cell-based therapy, and virus-mediated therapy, should be summarized. The authors explain the detail in manuscript, but it is better to overview the role of nanoparticles and make table.
Response: We agree with the suggestion of the reviewer. The content has been included in a table format (Table 2) in the revised manuscript as suggested.
- The citation number should be checked all. The reference number is reverse, for example page 5 ((14)(13), (16)(13)).
Response: We thank the reviewer for pointing out the citation error, which has now been corrected in the revised manuscript.
- Some abbreviation appear without spell out, for example page 4 (5 DNA), page 8 (HA). Also, abbreviation should be spell out at first appearance.
Response: We thank the reviewer for pointing out the missing information. The abbreviations have been checked and the full form spelt out at first mention in the revised manuscript as suggested.
I hope, we have satisfactorily answered all the comments of the reviewers. Please feel free to contact me for further clarification.
With best regards,
K. Uma Maheswari, Ph. D.
Reviewer 2 Report
This review elaborates on the research progress of nanomedicine used to improve the efficacy of tumor immunotherapy. It guides the further research of tumor immunotherapy which is helpful to a certain degree. We propose that this letter could be accepted when the following questions are responded to.
1. It would be better to list the tables in the Recent Developments in Cancer Immunotherapy section for clarity.
2. Authors should discuss the shortcomings of each nanoparticle or therapeutic strategy and corresponding solutions.
3. Nanoparticle-assisted tumor immunotherapy has shown good results in experiments, but so far, these treatments have been rarely used in clinical practice. Please discuss and compare this treatment strategy with current commonly used immunotherapy in clinical practice, such as PD-L1, PD-1, and CAR-T, etc.
4. More content should be added to the 'Conclusion and perspectives' section to elaborate on the application prospect of nanomedicines.
5. At the end of the first paragraph in the Introduction section, please use a sentence or two to state your literature-searching strategy and the overall objective of this review paper.
6. The author mentions in the abstract “The amenability of nanoparticles towards surface functionalization and tunable physicochemical properties, size, shape, and surfaces charge have been successfully harnessed for immunotherapy as well as combination therapy against cancer” Does the physical properties of nanoparticles, such as charge and shape, affect tumor immunity?
Author Response
Response to Reviewer-2 comments
- It would be better to list the tables in the Recent Developments in Cancer Immunotherapy section for clarity.
Response: We thank the reviewer for the suggestion. A separate table titled ‘Recent Developments in Cancer Immunotherapy’ (Table 2) has been included in the revised manuscript as suggested.
- Authors should discuss the shortcomings of each nanoparticle or therapeutic strategy and corresponding solutions.
Response: We thank the reviewer for the suggestion. Additional information on the limitations of each class of nanoparticles and therapeutic strategy has been included in the revised manuscript.
- Nanoparticle-assisted tumor immunotherapy has shown good results in experiments, but so far, these treatments have been rarely used in clinical practice. Please discuss and compare this treatment strategy with current commonly used immunotherapy in clinical practice, such as PD-L1, PD-1, and CAR-T, etc.
Response: We thank the reviewer for the suggestion. We have included a discussion on the nanoparticles-assisted treatment strategy with clinically used immunotherapy such as PD-L1, PD-1, and CAR-T in the revised manuscript as suggested.
- More content should be added to the 'Conclusion and perspectives' section to elaborate on the application prospect of nanomedicines.
Response: We thank the reviewer for the valuable suggestion. The conclusion and perspectives section has now been further elaborated in the revised manuscript as suggested.
- At the end of the first paragraph in the Introduction section, please use a sentence or two to state your literature-searching strategy and the overall objective of this review paper.
Response: We thank the reviewer for the suggestion. The overall objective of this review and the literature-search methodology has been included in the revised manuscript.
- The author mentions in the abstract “The amenability of nanoparticles towards surface functionalization and tunable physicochemical properties, size, shape, and surfaces charge have been successfully harnessed for immunotherapy as well as combination therapy against cancer” Does the physical properties of nanoparticles, such as charge and shape, affect tumor immunity?
Response: Drug release at the site of the tumour that is being targeted is aided by nanoparticle modification both in terms of functionalization and morphological features. The functionalization with specific ligands that interact with unique biomolecular signatures on the target cell surface can enhance site-specific action while surface modification of the nanoparticle with the antigen fragment can aid in selective activation of immune cells towards the cancer cells. Functionalization with poly(ethylene glycol) chains enhance the circulation time for sustained immune potentiating effect. Further, the size and shape of the nanoparticles aid in cellular uptake and tumour penetration, resulting in superior anti-tumour activity. Smaller size of nanoparticles below 300 nm are preferred for cell internalization while a spherical or rod-like shape is best suited for accumulation inside the tumour tissue. Thus, the attributes of nanoparticles have been exploited for immunotherapy.
I hope, we have satisfactorily answered all the comments of the reviewers. Please feel free to contact me for further clarification.
With best regards,
Uma Maheswari, Ph. D.
Reviewer 3 Report
In this manuscript, Gayathri et al. give a review om recent progress in nanotechnology for efficient immunotherapy. The review also highlights recent trends in immunotherapy strategies to be used independent as well as in combination for destruction of cancer cells as well as prevent metastasis and recurrence. This review is scientifically sound and well organized. It is recommended for publication after some minor revisions.
1. The annotation for each figure should be uniform in Capitalization or not Capitalization.
2. The authors should provide a singlet section to point out the challenges and limitations for the development of nano-immunotherapy.
Author Response
Response to Reviewer-3 comments
In this manuscript, Gayathri et al. give a review on recent progress in nanotechnology for efficient immunotherapy. The review also highlights recent trends in immunotherapy strategies to be used independent as well as in combination for destruction of cancer cells as well as prevent metastasis and recurrence. This review is scientifically sound and well organized. It is recommended for publication after some minor revisions.
- The annotation for each figure should be uniform in Capitalization or not Capitalization.
Response: We thank the reviewer for the suggestion. As per the suggestions, the annotation for each figure is checked and corrected for uniformity in the revised manuscript.
- The authors should provide a single section to point out the challenges and limitations for the development of nano-immunotherapy.
Response: We thank the reviewer for the suggestion. The challenges and limitations in the field of nano-immunotherapy have been included in the revised manuscript.
I hope, we have satisfactorily answered all the comments of the reviewers. Please feel free to contact me for further clarification.
With best regards,
Uma Maheswari, Ph. D.
Round 2
Reviewer 2 Report
The author has dealt with the shortcomings of the article very well, and the revised manuscript can be published, which will provide a reference for research.